# Ubiquitin ligation to F-box protein targets by SCF–RBR E3–E3 super-assembly

Daniel Horn-Ghetko[1], David T. Krist[1,5], J. Rajan Prabu[1], Kheewoong Baek[1], Monique P. C. Mulder[2], Maren Klügel[1], Daniel C. Scott[3], Huib Ovaa[2,6], Gary Kleiger[4] & Brenda A. Schulman[1,3✉]

E3 ligases are typically classified by hallmark domains such as RING and RBR, which are thought to specify unique catalytic mechanisms of ubiquitin transfer to recruited substrates[1,2]. However, rather than functioning individually, many neddylated cullin–RING E3 ligases (CRLs) and RBR-type E3 ligases in the ARIH family—which together account for nearly half of all ubiquitin ligases in humans—form E3–E3 super-assemblies[3–7]. Here, by studying CRLs in the SKP1–CUL1–F-box (SCF) family, we show how neddylated SCF ligases and ARIH1 (an RBR-type E3 ligase) co-evolved to ubiquitylate diverse substrates presented on various F-box proteins. We developed activity-based chemical probes that enabled cryo-electron microscopy visualization of steps in E3–E3 ubiquitylation, initiating with ubiquitin linked to the E2 enzyme UBE2L3, then transferred to the catalytic cysteine of ARIH1, and culminating in ubiquitin linkage to a substrate bound to the SCF E3 ligase. The E3–E3 mechanism places the ubiquitin-linked active site of ARIH1 adjacent to substrates bound to F-box proteins (for example, substrates with folded structures or limited length) that are incompatible with previously described conventional RING E3-only mechanisms. The versatile E3–E3 super-assembly may therefore underlie widespread ubiquitylation.

Virtually all eukaryotic cellular pathways (cell division, transcription, signalling and more) are regulated by protein ubiquitylation catalysed by E3 ligases, thought to function within E1–E2–E3 tri-enzyme cascades. Ten to twenty per cent of ubiquitylation in human cells is mediated by CRL E3s[8]. CRLs are multiprotein assemblies that contain a core 'catalytic' CUL–RBX (also known as cullin–RING) complex and an interchangeable substrate-specific receptor. Much of our understanding of CRLs comes from studies of the SCF family, in which different F-box proteins (about 70 in total in humans) act as receptors that recognize distinct substrates[9–14]. Different members of this E3 family are denoted as SCF combined with a superscript name of the specific F-box protein in the complex (for example, SCF[FBXW7] denotes an SCF complex in which the F-box protein is FBXW7). Substrate-bound complexes of SKP1 and the F-box protein bind interchangeably to the N-terminal end of the elongated CUL1 protein[15]. The C-terminal end of CUL1 forms an obligate multidomain assembly with RBX1, via an intermolecular C/R domain (also known as an α/β domain) that joins RBX1 with CUL1[15]. The E3 ligase RING domain of RBX1 and the C-terminal WHB domain of CUL1 are flexibly tethered to the C/R domain[16,17]. Covalent linkage of the ubiquitin-like protein NEDD8 to the WHB domain of CUL1 activates ubiquitylation[12–14,16–18]. In a given cellular condition, dozens of structurally distinct F-box proteins are incorporated into neddylated SCFs; the repertoire is determined in an intricate assembly pathway, regulated in part by substrates marked for ubiquitylation[19–27].

Neddylated SCF E3 ligases mediate ubiquitylation by the RING domain of RBX1 transiently recruiting a ubiquitin-carrying enzyme from which ubiquitin is transferred to substrates. As with other RING E3s[2,28–30], neddylated SCF ligases partner with several E2-type ubiquitin-carrying enzymes: typically, E2 enzymes in the UBE2D family directly ubiquitylate substrates, whereas those in the UBE2G1 and UBE2R families extend polyubiquitin chains[6,31]. However, knocking down or deleting these E2 enzymes individually stabilizes only some substrates of SCF ligases[4,6,31].

Notably, numerous cellular neddylated SCFs and some other CRLs have been reported to associate with ARIH1, an E3, to form 'E3–E3' ligases in which ARIH1 serves as the ubiquitin-carrying enzyme[3–5]. Mechanistically, ubiquitin is transferred from the ARIH1-bound E2 enzyme UBE2L3 to the catalytic cysteine of ARIH1, and then from ARIH1 to the substrate bound to the SCF E3 ligase[4]. Knocking down ARIH1 stabilizes substrates of several CRLs[4], including the paradigmatic phosphorylated p27 and cyclin E substrates of SCF[SKP2] and SCF[FBXW7], respectively. Moreover, CRISPR screens have shown that ARIH1 is essential for cell viability, similar to CUL1 and NEDD8[32,33]. Despite this importance, there are no structures that show how different types of E3 ligases function in a complex, and thus it is not possible to model E3–E3 ubiquitylation. Alone, ARIH1 is autoinhibited—its so-called 'Ariadne' domain intramolecularly blocks its catalytic 'Rcat' domain (also known as RING2)[34]—but it is unclear how ARIH1 is activated upon binding a NEDD8-modified CRL[3,4]. Despite several landmark structures[5,34–46], an entire cycle (from relief of autoinhibition, followed by ubiquitin transfer between catalytic cysteines of an E2 enzyme and that in the RBR E3 Rcat domain,

[1]Department of Molecular Machines and Signaling, Max Planck Institute of Biochemistry, Martinsried, Germany. [2]Oncode Institute, Department of Cell and Chemical Biology, Chemical Immunology, Leiden University Medical Centre, Leiden, The Netherlands. [3]Department of Structural Biology, St Jude Children's Research Hospital, Memphis, TN, USA. [4]Department of Chemistry and Biochemistry, University of Nevada, Las Vegas, Las Vegas, NV, USA. [5]Present address: Carle Illinois College of Medicine, Champaign, IL, USA. [6]Deceased: Huib Ovaa. ✉e-mail: schulman@biochem.mpg.de

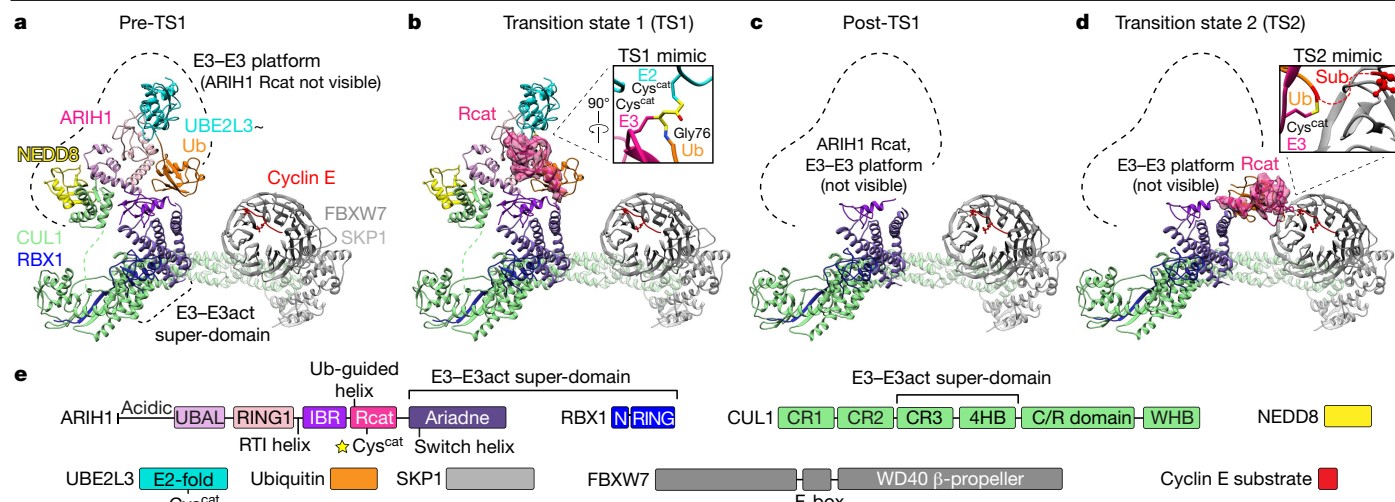

**Fig. 1 | Snapshots that represent intermediates during ARIH1-catalysed ubiquitylation of the phosphorylated cyclin E substrate of SCF^FBXW7.** Coordinates were refined against high-resolution cryo-EM data (TS1 and TS2), or modelled by fitting high-resolution structures into moderate-resolution maps (pre- and post-TS1). The Rcat domain of ARIH1 traverses more than 120 Å to relocate from its autoinhibited position, collect ubiquitin from UBE2L3 and then deliver ubiquitin to a substrate bound to the F-box protein. Sub, substrate. **a**, Pre-TS1. Neddylated SCF-activated ARIH1 binds thioester-bonded UBE2L3-ubiquitin (Ub), mimicked by ubiquitin isopeptide bonded to a mutant UBE2L3 in which the catalytic cysteine is substituted with lysine. **b**, TS1. Transfer of the C terminus of ubiquitin from UBE2L3 to the catalytic cysteine (Cys^cat) of ARIH1 (in the Rcat domain) is mimicked by simultaneously linking ubiquitin and the catalytic cysteines of UBE2L3 and neddylated SCF-activated

ARIH1. **c**, Post-TS1. The C terminus of ubiquitin thioester-bonded to the catalytic cysteine of ARIH1 was mimicked by neddylated SCF-dependent reaction of ARIH1 with ubiquitin–VME. In the major class from cryo-EM data, the only ARIH1 domain visible in the map is the Ariadne domain bound to CUL1–RBX1 in the E3–E3act super-domain. Much of ARIH1, its Rcat-linked ubiquitin, CUL1 WHB and NEDD8 are presumably dynamic. **d**, TS2. Transfer of ubiquitin from the catalytic cysteine of ARIH1 to the substrate bound to the F-box protein is mimicked by simultaneously linking ubiquitin, the catalytic cysteine of ARIH1 and the acceptor site on the cyclin E phosphopeptide. **e**, Guide to colouring of domains and proteins that participate in E3–E3-mediated ubiquitylation of substrates bound to an F-box protein. C/R, cullin–RBX; N, N terminus.

and subsequently to substrate) has not yet been visualized for any RBR-type E3 ligase—let alone for one transferring ubiquitin to a substrate recruited to a different E3 ligase.

## Visualizing ubiquitylation intermediates

To visualize assemblies that mediate neddylated SCF-dependent ubiquitin transfer from the E2 UBE2L3 to ARIH1 and then to a substrate bound to an F-box protein, we obtained cryo-electron microscopy (cryo-EM) data for chemically stable proxies for the two short-lived transition states (TS1 and TS2) and for intervening intermediates (pre-TS1 and post-TS1) (Extended Data Figs. 1–4, Extended Data Table 1, Supplementary Fig. 1). Studying ubiquitylation of substrate phosphopeptides derived from cyclin E and p27 by neddylated SCF^FBXW7 and SCF^SKP2, respectively, enabled us to visualize interchangeable F-box proteins and their different substrates: FBXW7 engages phosphorylated cyclin E via its β-propeller domain[47], whereas the concave leucine-rich repeat domain of SKP2 enwraps another protein (CKSHS1) that binds phosphorylated p27[48] (with or without supramolecular association with cyclin A–CDK2). However, the catalytic super-assemblies, which involve UBE2L3, ubiquitin, ARIH1, RBX1, the C-terminal regions of CUL1, and NEDD8, are superimposable for a given transition state for both substrate-bound SCF ligases. We compared the cryo-EM maps, which revealed a CUL1–RBX1–ARIH1 super-domain (which we term 'E3–E3act') that amalgamates the two E3 ligases and conformationally activates ubiquitylation throughout the multistep reaction (Fig. 1). Meanwhile, the catalytic Rcat domain of ARIH1 traverses more than 120 Å to relocate from its autoinhibited position, collect ubiquitin from UBE2L3 and then deliver ubiquitin to a substrate bound to an F-box protein (Fig. 1).

Before ubiquitin is transferred from the catalytic cysteine of UBE2L3 to the catalytic cysteine of ARIH1, neddylated SCF-activated ARIH1 binds the UBE2L3-ubiquitin conjugate (in which ~ refers to thioester linkage to a catalytic cysteine or a chemically stabilized mimic). We

visualized this pre-TS1 intermediate using a stable isopeptide-bonded mimic of the reactive thioester-bonded UBE2L3-ubiquitin conjugate (Fig. 1a, Extended Data Fig. 4a). Modelling high-resolution structures into a 4.5 Å-resolution cryo-EM map (specifically, a complex between UBE2L3-ubiquitin, ARIH1 and substrate-bound, neddylated SCF^FBXW7) showed that NEDD8 contributes to an intermolecular 'E3–E3 platform', which extends 70 Å perpendicularly from the E3–E3act super-domain. The E3–E3 platform presents the UBE2L3-ubiquitin conjugate for nucleophilic attack (Fig. 1a). ARIH1 is relieved from autoinhibition, its Rcat domain not visible and presumably mobile before the TS1 reaction.

To visualize the TS1 reaction (ubiquitin transfer from UBE2L3 to ARIH1), we installed an electrophile between the catalytic cysteine of UBE2L3 and the penultimate glycine of ubiquitin that captures the catalytic cysteine of ARIH1 depending on native reaction requirements (Extended Data Figs. 1, 2a–d). A 3.6 Å-resolution reconstruction largely resembles the pre-TS1 intermediate, but also shows the catalytic Rcat domain of ARIH1 and the preceding helix gripping the UBE2L3-linked ubiquitin (Fig. 1b, Extended Data Figs. 3c, 4b). To capture ubiquitin in the TS1 reaction, the catalytic cysteine of ARIH1 is relocated approximately 60 Å from its position in autoinhibited ARIH1.

In the post-TS1 intermediate, the C terminus of ubiquitin is liberated from UBE2L3 and is instead thioester-bonded to ARIH1. Cryo-EM data for a stable proxy, generated by neddylated SCF-dependent reaction of ARIH1 with the synthetic electrophilic probe ubiquitin–vinyl methyl ester (Ub–VME) (Extended Data Fig. 1d), were refined to three major classes. In all classes, the E3–E3act super-domain and much of the SCF are resolved. However, differences between the three classes suggest that NEDD8 and its covalently linked CUL1 domain, and the bulk of the ARIH1-ubiquitin conjugate, are mobile (Fig. 1c, Extended Data Fig. 4c).

In the TS2 reaction, ubiquitin is transferred from ARIH1 to an SCF-bound substrate. Dehydroalanine chemistry linking the C terminus of ubiquitin to an N-terminal acceptor site on a peptide substrate enabled capturing the catalytic cysteine of ARIH1 as a stable mimic of

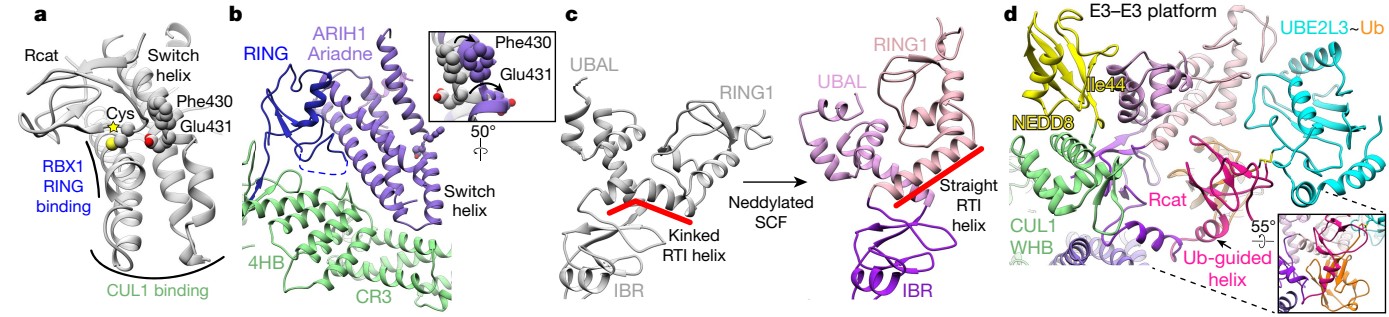

**Fig. 2 | Amalgamated intermolecular E3–E3act super-domain and E3–E3 platform activate neddylated SCF–ARIH1 ubiquitylation. a**, Autoinhibited ARIH1 (Protein Data Bank (PDB) code 4KBL[34]), showing switch helix residues of the Ariadne domain securing the Rcat domain. Yellow star labels the catalytic cysteine. RBX1- and CUL1-binding regions are indicated. **b**, In the E3–E3act super-domain, the Ariadne domain of ARIH1 (purple) binds CUL1 (green)–RBX1 (blue), accompanied by an activating bend-to-kink remodelling of the switch helix and ARIH1 side-chain relocation (inset). **c**, Side-by-side comparison of UBE2L3-ubiquitin-binding elements of ARIH1 in autoinhibited (left) (grey) (PDB 5UDH[49]) or TS1 (right) conformation (coloured as in Fig. 1e). Structures were aligned over the IBR domain. Upon binding to neddylated SCF E3 ligase, the RTI helix of ARIH1 is remodelled and other elements are rearranged. **d**, Intricate catalytic assembly for TS1. The E3–E3 platform, which includes CUL1-linked NEDD8, cradles UBE2L3-ubiquitin. Inset, the Ub-guided helix of ARIH1 binds the Ile44 patch of ubiquitin and guides the Rcat domain, which captures the C terminus of ubiquitin from UBE2L3. Colouring as in Fig. 1e.

the TS2 intermediate (Extended Data Fig. 2e–h). We obtained cryo-EM reconstructions of various TS2 assemblies, which superimposed for neddylated SCFs with wild-type ARIH1, and for unneddylated SCFs with an ARIH1 mutant (F430A/E431A/E503A, which relieves autoinhibition and allows ubiquitylation of substrates bound to unneddylated SCFs[4]) (Extended Data Fig. 4d, e, Extended Data Table 1). The highest-resolution (3.6 Å) map, obtained using the ARIH1 mutant, permitted unambiguous docking of catalytic elements—including ubiquitin linked to the ARIH1 Rcat domain, which was translocated approximately 60 Å from the TS1 active site to confront the F-box-protein-bound substrate (Fig. 1d, Extended Data Fig. 3f). The E3–E3 platform was poorly resolved: its dismantling potentially enabled this TS2 configuration.

## Allosteric release from autoinhibition

The E3–E3act super-domain adjoins the Ariadne domain of ARIH1 with a composite surface from the RING domain of RBX1 and the 'CR3' and '4HB' domains of CUL1. This depends on conformational changes from both E3 ligases (Fig. 2a, b, Extended Data Fig. 5a, b). In a neddylated SCF without a recruited ubiquitin-carrying enzyme, the RING domain of RBX1 is relatively mobile[16,17]. However, the RING domain of RBX1 adopts a specific orientation to bind ARIH1 (Fig. 2b). Meanwhile, the Ariadne domain of ARIH1 is twisted relative to its autoinhibitory conformation[34,43,49]. Binding to CUL1–RBX1 is accompanied by a 165°-to-125° bend-to-kink transition between Phe430 and Glu431, in what we term the Ariadne 'switch-helix'. Glu431 makes an about-face 12 Å translation accompanied by a 4 Å outward shifting of Phe430 and propagated relocation of flanking residues. This contrasts with autoinhibited ARIH1, in which these side chains face into the Ariadne domain to secure intramolecular interactions with the Rcat domain and preceding meandering loop (Fig. 2a, b). This switch-helix kink thus releases the Rcat domain, which triggers E3–E3 ligase activity. In addition, with the Ariadne domain of ARIH1 straddling the distal end of CUL1–RBX1, the E3–E3act domain ensures that each ubiquitin transfer reaction catalysed by the RBR domain couples to a requisite SCF feature—neddylation and substrate recruitment (Fig. 1a–d).

## Amalgamated E3–E3 ligase TS1 reaction

Neddylation stimulates ARIH1 binding to endogenous cellular CRLs[3–5], and the first E3–E3 transition state[4]. The cryo-EM data show that NEDD8 forms part of the E3–E3 platform displaying the UBE2L3-ubiquitin conjugate (Fig. 1a, b).

Both E3 ligases undergo conformational changes to form the E3–E3 platform (Extended Data Fig. 5c, d). In the absence of other factors, NEDD8 and its covalently linked WHB domain of CUL1 are mobile relative to the rest of the SCF[17]. However, in the E3–E3 platform, the WHB domain of CUL1 and NEDD8 pack against each other, and the Ile44 patch of NEDD8 engages the UBAL subdomain of ARIH1. In the resultant composite platform, the orientations of the ARIH1 UBAL domain and RING1, RTI helix and IBR elements are remodelled (Fig. 2c, d). Compared to autoinhibited ARIH1[34,43,49], the straightening of the RTI helix and rotation of the UBAL domain towards CUL1 accommodates canonical RBR catalytic presentation of the UBE2L3-ubiquitin conjugate[41].

We next asked how the Rcat domain of ARIH1 is positioned to receive ubiquitin from UBE2L3. In the pre-TS1 intermediate, the Rcat domain of ARIH1 is not visible: it is presumably mobilized upon release from Ariadne-domain-mediated autoinhibition (Fig. 1a). Nonetheless, the Rcat domain is physically connected via linkers to the IBR and Ariadne domains, which are on opposite sides of the E3–E3act super-domain. In the TS1 intermediate, the linker that connects the ARIH1 IBR and Rcat domains—which is displaced by the switch-helix conformational change—is remodelled into a Ub-guided helix that synergizes with the Rcat domain to promote its capturing of the UBE2L3-linked ubiquitin (Figs. 1b, 2d). The Ub-guided helix is rotated nearly 180° to bind the Ile44-centred hydrophobic patch of ubiquitin. The Ub-guided helix thus projects the ensuing Rcat domain towards ubiquitin (Extended Data Fig. 5e). Catalytically important residues of the Rcat domain grip the extended C-terminal tail of ubiquitin in a β-sheet, which culminates in linkage between ubiquitin and ARIH1 (Extended Data Fig. 6, Supplementary Table 1).

## RBR ubiquitin transfer to SCF substrates

The arrangement of ubiquitin bound to the ARIH1 Ub-guided helix and Rcat domain is preserved between E3–E3 TS1 and TS2 intermediates (Figs. 2d, 3a). A parallel arrangement was previously observed for HOIP[41] (Extended Data Fig. 5f–h). We thus define these E3-ubiquitin elements as the 'RBR ubiquitin transferase module'. For ARIH1-linked ubiquitin to approach a substrate bound to an F-box protein, the ubiquitin transferase module undergoes an approximately 100° rotation about the E3–E3act super-domain after TS1 (Fig. 3b).

The Ub-guided helix has a crucial role in positioning the active site. This sequence contributes to securing the Rcat and Ariadne domains in autoinhibited ARIH1[34], and forms the Ub-guided helix extending from connections to the E3–E3act super-domain for the TS1 reaction. In the

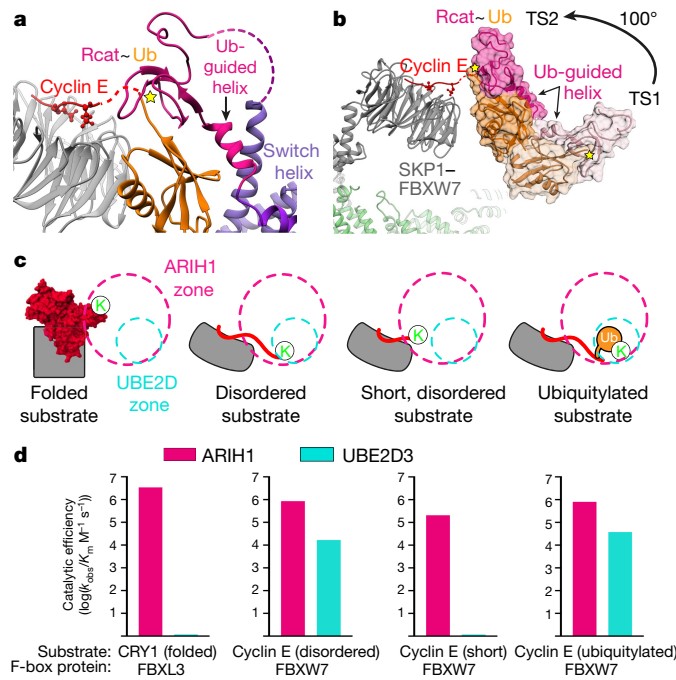

**Fig. 3 | ARIH1 ubiquitylation of a range of substrates recruited to diverse F-box proteins. a**, Structure representing TS2: ubiquitin transfer from ARIH1 to SCF substrate. The Ub-guided helix of ARIH1 and ensuing Rcat domain bound to ubiquitin form a ubiquitin transferase module barricaded by the switch helix of ARIH1. **b**, Approximately 100° reorientation of ubiquitin transferase module (shown in surface) between TS1 (light pink and melon) and TS2 (dark pink and orange), shown by aligning CUL1–RBX1 and the E3–E3act super-domain of the two transition state structures. Yellow star denotes the catalytic cysteine of ARIH1. **c**, Cartoon representations of various F-box proteins (grey) and their substrates (red) relative to zones accessible to ubiquitin-linked active sites of ARIH1 (pink) and UBE2D (cyan), based on structural modelling. K, substrate lysine. **d**, Graphs showing the mean value of catalytic efficiencies ($k_{obs}/K_m$) for ARIH1 (pink)- and UBE2D3 (cyan)-mediated ubiquitylation with indicated substrate and F-box protein, as depicted in **c**. $n = 3$ independent experiments.

### ARIH1 E3 targets diverse SCF substrates

The cryo-EM maps that represent the TS2 intermediates of SCF^FBXW7 and SCF^SKP2 show the ARIH1-ubiquitin active site adjacent to disordered substrates recruited to distinct F-box proteins (Extended Data Fig. 4d). To gain insights into E3–E3 ubiquitylation of structurally diverse substrates, we generated models based on the TS2 structures. The E3–E3

mechanism appears structurally compatible with a folded substrate, disordered substrates proximal to various F-box proteins, and a ubiquitylated substrate (Fig. 3c, Extended Data Fig. 7). Given that neddylated SCF E3s can use a range of ubiquitin-carrying enzymes (E2s as well as ARIH1[4,13,14]), we also generated models based on the previous structure representing ubiquitin transfer from a UBE2D-family E2 enzyme to a peptide substrate of neddylated SCF^β-TRCP (ref.[17]). The placement of UBE2D appears structurally optimal for substrates recruited to the F-box protein β-TRCP, and also seems compatible with substrates of some other neddylated SCFs.

We quantitatively compared the E3–E3 and E2–E3 mechanisms, and determined kinetic parameters by rapid quench-flow methods (Fig. 3d, Extended Data Fig. 8, Extended Data Table 2). The relatively similar catalytic efficiencies for ARIH1 with all tested neddylated SCFs and substrates confirmed the broad utility of the E3–E3 mechanism[4]. The conventional E2–E3 mechanism (with UBE2D3) was optimal for neddylated SCF^β-TRCP. UBE2D3 could also ubiquitylate disordered peptide substrates of sufficient length to simultaneously engage the F-box protein and the active site of this E2. However, ubiquitylation of substrates that cannot accommodate the structurally modelled E2–E3 catalytic geometry—with folded structure (CRY1 recruited to the F-box protein FBXL3) or with a limited distance between the F-box-protein-binding degron and the acceptor lysine ('short' truncated versions of cyclin E or β-catenin recruited to FBXW7 or β-TRCP, respectively)—were quantifiable only for ARIH1 (Extended Data Fig. 8, Extended Data Table 2).

The structural comparison also showed the same surfaces of CUL1-linked NEDD8 and the RING domain of RBX1 engaging ARIH1 and UBE2D, albeit in different relative orientations (Extended Data Fig. 5a–d). Also, the ARIH1-binding site on RBX1 reportedly interacts with a UBE2R E2 enzyme[50]. Accordingly, our stable proxy for the E3–E3 post-TS1 intermediate (which represents ARIH1 actively engaged by a neddylated SCF ligase) excluded ubiquitylation by UBE2D3 or UBE2R2 (Extended Data Fig. 7f, g). The ubiquitylation reactions using UBE2D3 and UBE2R2 were unaffected by a catalytically inactive version of ARIH1. The results suggest that when a neddylated SCF is mediating ubiquitylation with ARIH1, other ubiquitin-carrying enzymes would be blocked. However, after a substrate is modified, ubiquitin-free ARIH1 could disengage, allowing the neddylated SCF E3 to use a different ubiquitin-carrying enzyme—presumably one with superior kinetic properties for modifying the particular F-box protein client.

## Discussion

This work defines how transient amalgamation of two types of E3 ligase, each inactive on its own, into a neddylated SCF–ARIH1 E3–E3 super-assembly achieves substrate ubiquitylation. Radically different configurations for the two transition states depend on both E3 ligases sculpting each other into catalytic conformations unattainable by either E3 ligase alone. The structures thus reveal how two E3 ligases (neddylated SCFs and ARIH1) co-evolved specifically to mediate ubiquitylation with each other.

The structures—together with previous studies[4]—suggest that neddylated SCF–ARIH1 ubiquitylation proceeds in a cycle, choreographed by cellular regulatory cues and by the successively changing C-terminal linkage of ubiquitin (Fig. 4). After neddylation of a substrate-bound SCF E3, the RING domain of RBX1 and NEDD8 could co-contact an autoinhibited ARIH1–UBE2L3-ubiquitin complex. The Ariadne domain of ARIH1 twists upon binding CUL1–RBX1. This kinks the ARIH1 switch helix, thus unleashing the Rcat domain. Concomitantly, NEDD8, its linked CUL1 and ARIH1 elements form an E3–E3 platform that presents the UBE2L3-ubiquitin conjugate. Next, the UBE2L3-linked ubiquitin would lure the linker between the IBR and Rcat domains of ARIH1, and promote remodelling of the linker into the Ub-guided helix, which directs the Rcat domain to engage the C-terminal tail of ubiquitin. The

TS2 intermediate (Figs. 1b, 3a, b, Extended Data Fig. 5e), the Ub-guided helix is reoriented back towards the Ariadne domain, but its trajectory is limited: the ubiquitin-bound helical structure precludes the conformation that is required for autoinhibition. The switch helix of the E3–E3act super-domain appears to barricade the position of the ubiquitin transferase module. As a result, the ubiquitin transferase module is restricted—but not fixed—within a zone that directs the active site towards the F-box-protein-bound substrate (Fig. 3a, Extended Data Figs. 3d–f, 4d, 7). Notably, the cryo-EM data for TS2 intermediates refine to several classes with slight variations in the relative positions of the F-box protein and the ubiquitin transferase module. The resolution of the ubiquitin transferase module is lower than for the adjacent E3–E3act super-domain, which potentially reflects relative mobility. Mutational results are consistent with the crucial role of the Ub-guided helix and with its restraint by the E3–E3act super-domain contributing to the TS2 reaction[4] (Extended Data Fig. 6c–f, Supplementary Table 1).

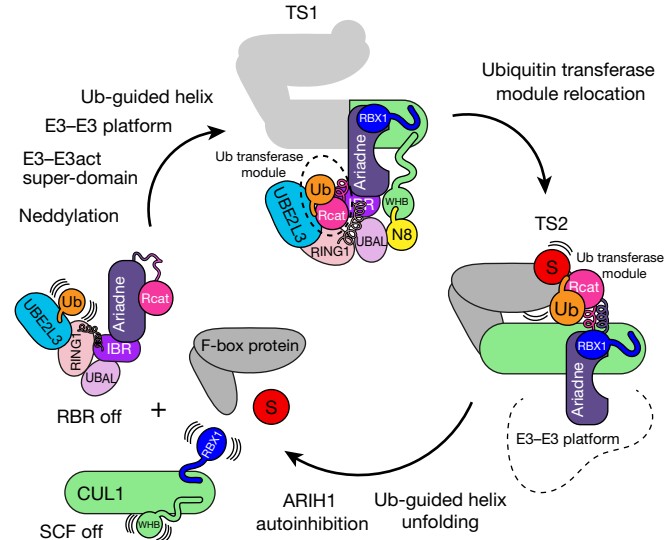

**Fig. 4 | Proposed transient amalgamation–ubiquitylation–disassembly cycle of SCF–ARIH1 RBR-type E3–E3 ligase.** Model for neddylated SCF–ARIH1 E3–E3 ubiquitylation. Portions of proteins that contribute to the transition states are coloured as in Fig. 1e. (1) ARIH1 is autoinhibited, and SCF is unassembled and unneddylated. (2) NEDD8 linkage persists on substrate-bound SCF E3 ligase, enabling amalgamation with ARIH1 via the E3–E3act super-domain and E3–E3 platform. The switch helix of the Ariadne domain of ARIH1 is twisted, and the Rcat domain of ARIH1 is freed from autoinhibition. (3) In TS1, the Ariadne domain of ARIH1 binds CUL1–RBX1 in the E3–E3act super-domain, and NEDD8-bound ARIH1 elements are remodelled in the E3–E3 platform. The E3–E3 platform displays the linked ubiquitin of UBE2L3, which lures the Ub-guided helix of ARIH1 to promote the capture of UBE2L3-linked ubiquitin by the catalytic cysteine of ARIH1. The Ub-guided helix and Rcat domains of ARIH1, and bound ubiquitin, form a ubiquitin transferase module. (4) In TS2, the ubiquitin transferase module has undergone an approximately 100° translocation to deliver ubiquitin to the substrate bound to the F-box protein. (5) Without bound ubiquitin, the Ub-guided helix of ARIH1 could resume an autoinhibitory conformation. E3–E3 disassembles.

Ub-guided helix and Rcat domain of ARIH1, together with ubiquitin, form the ubiquitin transferase module. Upon the TS1 reaction, the C-terminal linkage of ubiquitin with UBE2L3 is severed and transferred to the catalytic cysteine of ARIH1, allowing relocation of the ubiquitin transferase module from the outskirts back towards the Ariadne domain. This would place the reactive ARIH1-ubiquitin active site near the substrate bound to the F-box protein, driving ubiquitylation of the substrate. Elimination of the thioester linkage between ARIH1 and ubiquitin would enable the Ub-guided helix to unfold and, together with the Rcat domain, to recapture the Ariadne domain. Resuming this autoinhibited conformation would reset the switch helix of ARIH1, promote E3–E3 dissociation and thus enable the neddylated SCF E3 to engage a different ubiquitin-carrying enzyme to polyubiquitylate the ubiquitin-primed substrate.

The 'mix-and-match' neddylated CRL system creates hundreds of distinct E3 ligases that recruit numerous, structurally diverse substrates. Although conventional ubiquitylation with the E2 UBE2D is optimal for some substrates recruited to particular neddylated SCFs, it does not readily accommodate others. By contrast, our data show that the E3–E3 structural mechanism places the catalytic centre in a zone that matches different F-box proteins, and comparable catalytic efficiencies for ARIH1 ubiquitylation of their various substrates (Fig. 3, Extended Data Table 2). Thus, the E3–E3 structures redefine the list of parts in modular CRLs: just as interchangeable F-box proteins recruit distinct substrates, mix-and-match ubiquitin-carrying enzymes determine the ubiquitylation of these diverse substrates.

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

# Methods

No statistical methods were used to predetermine sample size. The experiments were not randomized, and investigators were not blinded to allocation during experiments and outcome assessment.

### Cloning, protein expression and purification
Expression constructs were prepared and verified using standard molecular biology techniques. Coding sequences for all proteins are of human origin. Mutant versions of ARIH1, UBE2L3, NEDD8 and ubiquitin were generated using Quikchange system (Agilent).

GST–TEV–RBX1 (residue 5 to C terminus), full-length CUL1, GST–TEV–UBA1, SKP1, His–TEV–β-TRCP2, GST–TEV–FBXL3 and cryptochrome 1 (CRY1, residues 1–532) were expressed in *Trichoplusia ni* High-Five insect cells, with CUL1 and RBX1, SKP1 and β-TRCP2, or SKP1, FBXL3 and CRY1 co-expressed as previously described[4,17]. Proteins were purified by GST affinity chromatography followed by overnight TEV cleavage. Purification was completed by ion-exchange and size-exclusion chromatography. The neddylation components UBE2M and APPBP1–UBA3 were expressed in either *Escherichia coli* Rosetta (DE3) or BL21 Gold (DE3) cells as GST–thrombin fusion proteins. Fusion proteins were subjected to overnight protease cleavage and further purified by ion-exchange and size-exclusion chromatography. SKP1–FBXW7(ΔD) (a monomeric version of FBXW7 comprising residue 263 to the C terminus)[47], SKP1–SKP2 (residues 101 to the C terminus), SKP1–SKP2 (full length) and the p27 N-terminal domain (residues 22–106) were expressed in *E. coli* Rosetta 2 (DE3) or BL21 Gold (DE3) and purified as described for neddylation components, but with overnight TEV cleavage after affinity purification. Neddylation of CUL1–RBX1 along with fluorescent labelling of ubiquitin were carried out as previously described[51]. UBE2M(Y130L) was used to modify CUL1–RBX1 with the I44A mutant of NEDD8[51]. ARIH1 (and mutant derivatives), UBE2D3, UBE2R2 and UBE2L3 (wild type, C86K and C17A/C137A) were expressed in *E. coli* Rosetta 2 (DE3) and purified by GST affinity chromatography. Overnight TEV cleavage was carried out on-column followed by size-exclusion chromatography of cleavage reactions. Expression and purification of cyclin A(170 to C terminus)–CDK2, NEDD8 and CKSHS1 (5–73) were performed as previously described[4].

### Peptides
All peptides used for biochemical assays or activity-based probes were obtained from Max Planck Institute of Biochemistry Core Facility, New England Peptides or synthesized in the laboratory of H.O., and determined to be more than 95% pure after high-performance liquid chromatography (HPLC) purification.

The following peptides were bound via their phospho-degrons to F-box proteins, and represent substrates in cryo-EM structures. For pre-TS1 and TS1 (neddylated SCF-mediated ubiquitin transfer from UBE2L3 to ARIH1), the portion of the complex corresponding to the substrate for TS1 of neddylated SCF$^{SKP2}$ is: KRANRTEENVSDGSPNAGSVEQ(pT)PRRPGLRRRQTDYKDDDDK. For the pre-TS1 and TS1 for neddylated SCF$^{FBXW7}$, the portion of complex corresponding to cyclin E substrate is: KAMLSEQNRASPLPSGLL(pT)PPQ(pS)GKKQSSDYKDDDDK.

The following peptides were used as substrate portions of the TS2 activity-based probes (ABPs): TS2 p27 ABP, CNKRANRTEENVSDGSPNAGSVEQ(pT)PRRPGLRRRQTDYKDDDDK; TS2 cyclin E ABP, CKKAMLSEQNRASPLPSGLL(pT)PPQ(pS)GKKQSSDYKDDDDK. The peptide for TS2 cyclin E ABP alternative was K*AMLSEQNRASPLPSGLL(pT)PPQ(pS)GRRASY, in which K* is Dab(Alloc).

To monitor the transfer of fluorescent ubiquitin from either UBE2L3 (via ARIH1), UBE2D3 or UBE2R2 to SCF substrates, the following peptides were used as substrate in the ubiquitylation assays: cyclin E, KAMLSEQNRASPLPSGLL(pT)PPQ(pS)GRRQSS and 'sortase-able' cyclin E GGGGLPSGLL(pT)PPQ(pS)GKKQSSDYKDDDDK.

To monitor the transfer of ubiquitin from UBE2L3 or UBE2D3 to radiolabelled SCF substrates, the following peptides were used as substrate in the ubiquitylation assays: cyclin E, KAMLSEQNRASPLPSGLL(pT)PPQ(pS)GRRASY; cyclin E (short), KAGLL(pT)PPQ(pS)GRRASY; sortase-able cyclin E, GGGGGPLPAGLL(pT)PPQ(pS)GRRASY; β-catenin, KAAVSHWQQQSYLD(pS)GIH(pS)GATTAPRRASY; and β-catenin (short), KAYLD(pS)GIH(pS)GAGAGAPRRASY-OH.

### Generation of a ubiquitylated substrate for polyubiquitylation assays.
To generate ubiquitin-primed substrate, ubiquitin with a C-terminal LPETGG sequence was linked to a GGGG–cyclin E phosphopeptide by sortase-mediated transpeptidation as previously described[17].

### Generation of ABPs
**Preparation of His–Ub(1–75)–MESNa.** Most of the ABPs depended on preparation of His–Ub(1–75)–MESNa. Here, a mutant version of ubiquitin lacking its C-terminal glycine 76 was used to generate stable proxies for TS1 and TS2 intermediates that best mimicked atomic distances and geometry. N-terminal His-tagging was important for purifying Ub–MESNa and conjugated ABP products. First, His–Ub(1–75)–MESNa was cloned and prepared similarly as previously described[17,52], with protein first expressed in *E. coli* Rosetta 2 (DE3), cells resuspended and lysed in 20 mM HEPES pH 6.8, 50 mM NaOAc, 100 mM NaCl and 2.5 mM PMSF, and purified by Ni-NTA affinity chromatography. Next, the chitin-binding domain fused to ubiquitin was cleaved by diluting 5:1 (v/v) with 20 mM HEPES pH 6.8, 50 mM NaOAc, 100 mM NaCl and 100 mM MESNa. After overnight incubation at room temperature on a roller, the His–Ub(1–75)–MESNa product was purified via size-exclusion chromatography. It is critical to perform this purification in a pH range from 6 to 7 as higher pH promotes the hydrolysis of Ub–MESNa and consequent elimination of reactivity. His–Ub(1–75)–MESNa was analysed for its integrity versus level of hydrolysis via liquid chromatography coupled to mass spectrometry (LC–MS) to calculate appropriate amounts of starting material before ABP generation.

### Formation of TS1 ABP, to visualize assemblies representing TS1.
We exploited the dependence of ABPs on native enzyme activity[53] to synthesize a TS1 ABP 'in situ', capturing ubiquitin transfer from the catalytic cysteine of an E2 enzyme (here UBE2L3) to the catalytic cysteine of the neddylated SCF-activated RBR E3 ligase (here ARIH1). Because the catalytic cysteine of ARIH1 is occluded by its autoinhibitory Ariadne domain even when bound to UBE2L3[34,49], the formation of the ABP product occurs only when ARIH1 is activated by binding to a neddylated SCF E3 ligase[3]. The basis for the formation of the TS1 ABP–UBE2L3-Ub was coupling His–Ub(1–75)–MESNa to (E)-3-[2-(bromomethyl)-1,3-dioxolan-2-yl]prop-2-en-1-amine (BmDPA) (>95% purity, ChiroBlock) to yield a reactive species[4]. His–Ub(1–75)–MESNa (3 mg ml$^{-1}$) was mixed in a 5-ml Falcon tube with 0.4 M of BmDPA and 1 mM N-hydroxysuccinimide in 25 mM HEPES pH 6.7 and 100 mM NaCl. The reaction mix was incubated at 90 rpm at room temperature overnight. After desalting into 25 mM HEPES pH 6.7 and 100 mM NaCl, products were checked for successful conversion via LC–MS. Deprotection was performed by addition of 0.04 M p-TsOH (dissolved in 54% TFA (v/v)) to 0.5 mg ml$^{-1}$ reactive ubiquitin species and incubated for 1 h at room temperature. To eliminate TFA, the reaction mixture was precipitated and washed several times with cold ether (20-fold reaction volume). Protein flakes were air-dried and resuspended in 100 mM Na$_2$HPO$_4$ pH 6.0, 500 mM NaCl and 8 M urea. Ubiquitin was then refolded via dialysis in 20 mM Na$_2$HPO$_4$ and 100 mM NaCl, pH 6.0 overnight at 4 °C. The refolded, deprotected ubiquitin species (0.5 mg ml$^{-1}$ final concentration) was mixed with 5-fold excess UBE2L3 (C17A, C137A), incubated for 2 h at 30 °C and purified via nickel-affinity as well as size-exclusion chromatography, yielding a TS1 ABP–UBE2L3-ubiquitin. All cysteines except the catalytic cysteine of UBE2L3 were mutated to alanine as they would interfere with the conjugation to the deprotected ubiquitin species.

**Generation of ubiquitin–VME.** The Ub(1–75) peptide, bearing a free N terminus and its side chains protected, was synthesized on a trityl resin by Fmoc solid-phase peptide synthesis procedures (25 μmol scale) and removed from the resin using 1,1,1,3,3,3-hexafluoropropan-2-ol (HFIP) as previously described[54]. Gly-VME (10 equivalents) was coupled to the C terminus of Ub by using PyBOP (5 equivalents), triethylamine (Et₃N) (20 equivalents) in DCM (5 ml) and stirred for 16 h at ambient temperature. Excess Gly–VME was removed by washing the DCM solution with 1 M KHSO₄. The organic layer was dried with Na₂SO₄ and concentrated to dryness in vacuo. To remove the side-chain protecting groups, the residue was taken up in trifluoroacetic acid/triisopropylsilane/water (5 ml; 95/2.5/2.5) and stirred for 3 h at ambient temperature. The reaction mixture was added to a falcon tube containing ice-cold pentane/diethyl ether (1/3; 40 ml), upon which the product precipitated. The precipitate was isolated by centrifugation (1,500g, 6 min, 4 °C) and washed by three cycles of resuspension in ice-cold diethyl ether and centrifugation. Finally, the pellet was taken up in water/acetonitrile/acetic acid (65/25/10), frozen and lyophilized. Product was purified by prep-HPLC.

**Generation of TS2 ABPs to visualize TS2.** *TS2 ABPs used for high-resolution cryo EM and biochemical studies.* ARIH1 ubiquitylates a plethora of SCF substrates[4]. After ubiquitin is transferred from UBE2L3 to the catalytic cysteine of ARIH1, ARIH1 delivers ubiquitin to the SCF-recruited substrate. To understand how ubiquitin is transferred to SCF substrates by ARIH1, we generated peptide substrate–ubiquitin fusions and installed a central chemical handle to target the catalytic cysteine of ARIH1. To do this, native chemical ligation was used to couple His–Ub(1–75)–MESNa to a substrate phosphopeptide (p27 or cyclin E) equipped with an N-terminal cysteine. Taking the hydrolysed proportion of His–Ub(1–75)–MESNa into consideration, His–Ub(1–75)–MESNa (200 μM final concentration) and substrate peptide (1,000 μM final concentration) were mixed and incubated as previously described[17]. After native chemical ligation, the product was purified by Ni-NTA affinity and size-exclusion chromatography (final buffer was 50 mM NaPO₄ pH 8.0 and 50 mM NaCl). The cysteine in the Ub(1–75)–Cys–peptide product was then chemically converted with 2,5-dibromohexanediamide to dehydroalanine, yielding an ABP[55,56]. To achieve this, tris(2-carboxyethyl)phosphine (1 mM final concentration) was added to Ub(1–75)–Cys–peptide (700 μl at 300 μM) and incubated for 20 min at room temperature. After desalting (Zeba spin column, 2 ml, 7,000 molecular weight cut-off filter, Thermo Fisher) 2,5-dibromohexanediamide (14 mM final concentration, stock solution dissolved in DMSO) and Ub(1–75)–Cys–peptide were combined in a 2-ml tube in 50 mM NaPO₄ pH 8.0. The reaction mix was first incubated for 30 min at 23 °C with rocking at 90 rpm and then at 37 °C for 2 h. Finally, the reaction mix was desalted into 50 mM NaPO4 pH 8.0 (Zeba spin column, 5 ml, 7,000 molecular weight cut-off filter, Thermo Fisher).

*TS2 ABP alternative.* The starting point for the generation of the TS2 ABP alternative was the preparation of Ub(1–75)–Set and the substrate cyclin E phosphopeptide via a solid-phase peptide synthesis (SPPS) methodology adapted from a previous publication[57]. In brief, SPPS was performed on a Syro II MultiSyntech Automated Peptide synthesizer using standard 9-fluorenylmethoxycarbonyl (Fmoc)-based solid-phase peptide chemistry. Ub(1–75)–Set was synthesized as previously described[57] and the cyclin E sequence K*AMLSEQNRASPLPSGLL(pT)PPQ(pS)GRRASY with a Lys (K*) to Dab(Alloc) mutation was synthesized on a Fmoc-Y preloaded Wang resin. The resin was treated with Pd(PPh₃)₄ and Ph₃SiH to liberate the side chain N terminus, after which 4-((tert-butoxycarbonyl)amino)-3-(tert-butyldisulfanyl)butanoic acid was coupled. The cyclin E substrate peptide was deprotected and liberated from the resin by treatment with trifluoroacetic acid/triisopropylsilane/water (5 ml; 95/2.5/2.5) for 3 h at ambient temperature. The reaction mixture was added to a falcon tube containing ice-cold pentane/diethyl ether (1/3; 40 ml), upon which the product precipitated. The precipitate was isolated by centrifugation and washed in ice-cold diethyl ether (3×). Finally, the pellet was taken up in water/acetonitrile/acetic acid (65/25/10), frozen and lyophilized. The substrate peptide was purified by prep-HPLC. Subsequently, the substrate peptide and Ub(1–75)–Set were mixed and incubated as previously described[57]. After native chemical ligation, product was purified by prep-HPLC and treated with 2,5-dibromohexanediamide, yielding the TS2 ABP alternative. Product was purified by prep-HPLC and size-exclusion chromatography.

## Enyzme kinetics

**Rapid-quench flow kinetics to investigate the E3–E3 ubiquitin transfer mechanism.** Fluorescent ubiquitin was thioester-linked to UBE2L3 in a 'pulse' reaction incubating 10 μM UBE2L3 (also known as UBCH7), 15 μM fluorescent ubiquitin and 400 nM UBA1 in 25 mM HEPES, 150 mM NaCl, 2.5 mM MgCl₂, 1 mM ATP, pH 7.5 for 13 min at room temperature. The pulse reaction was quenched with 25 mM HEPES, 150 mM NaCl, 50 mM EDTA, pH 7.5 such that the final concentration of UBE2L3-ubiquitin was 800 nM, and incubated on ice for 5 min. Rapid-quench flow experiments were performed at room temperature by placing 800 nM UBE2L3-ubiquitin in syringe one and 600 nM ARIH1, 800 nM NEDD8–CUL1–RBX, 800 nM SKP1–FBXW7(ΔD), with or without 4 μM cyclin E in 25 mM HEPES, 150 mM NaCl, 0.5 mg ml⁻¹ BSA, pH 7.5 in syringe two. Syringes were mixed and quenched with 2× SDS–sample buffer at the indicated times. Reaction products were separated on 4–12% Bis-Tris gels, scanned on a Typhoon imager and quantified using ImageQuant software.

**Estimating the $K_m$ of ARIH1 and UBE2D for various SCF–substrate complexes.** Single-encounter reactions between ³²P-labelled substrate and SCF ligase were assembled in a reaction buffer containing 30 mM Tris, pH 7.5, 100 mM NaCl, 5 mM MgCl₂, 2 mM ATP and 1 mM DTT. For cyclin E, ubiquitylated cyclin E and cyclin E (short) peptide substrates and ARIH1, stock solutions of E1, ubiquitin and UBE2L3 were prepared (tube 2) and incubated for 5 min at room temperature before the addition of cold competitor substrate (for ubiquitylated cyclin E, competitor substrate was free peptide without sortase-linked ubiquitin). In a separate tube, neddylated CUL1–RBX1, SKP1–FBXW7(ΔD) and ³²P-labelled substrate were incubated for 5 min (tube 1) and then equally distributed to Eppendorf tubes. A twofold dilution series of ARIH1 was prepared, followed by the addition of an equal volume from the dilution series to each Eppendorf tube containing the SCF-labelled substrate complex. Reactions were initiated by adding an equal volume from tube 2 to each Eppendorf tube. After 10 s, reactions were quenched with 2× SDS–PAGE loading buffer containing 100 mM Tris-HCl, pH 6.8, 20% glycerol, 30 mM EDTA, 4% SDS and 0.02% bromophenol blue. For cyclin E, ubiquitylated cyclin E and cyclin E (short) peptide substrates and UBE2D, stock solutions of E1 and ubiquitin were prepared and then distributed to Eppendorf tubes. UBE2D3 aliquots from a twofold dilution series were then added to each tube followed by the addition of cold competitor substrate. Ubiquitylation reactions were initiated by adding equal volumes of SCF-labelled substrate from tube 1 that had been prepared as described previously in this section. For β-catenin and β-catenin (short) peptide substrates and ARIH1, stock solutions of E1, ubiquitin and UBE2L3 were prepared (tube 2) and incubated for 5 min at room temperature before distributing equal volumes to Eppendorf tubes. The ARIH1 dilution series was then added to each tube followed by the addition of cold competitor substrate. The reactions were initiated by the addition of equal volumes of SCF-labelled substrate mixture (now containing SKP1–β-TRCP2) from tube 1, incubated for 10 s, and quenched as described previously in this section. For β-catenin and β-catenin (short) peptide substrates and UBE2D3, the procedure was identical to that for cyclin E except that the SCF complex contained

SKP1–β-TRCP2. For ARIH1 or UBE2D3 and CRY1 substrate (bound to SKP1–FBXL3), $^{32}$P-labelled human ubiquitin was used to detect product formation. As CRY1 dissociation from SCF is thought to not occur over the time course, these reactions are also considered single encounter as regards substrate and SCF. For all reactions, substrates and products were separated on 18% SDS–PAGE gels followed by autoradiography and the quantification of unmodified substrate or products containing at least one or more ubiquitin (Image Quant TL software; GE). The fraction of product formed was plotted as a function of the UBE2D3 or ARIH1 concentration and fit to the Michaelis–Menten model (GraphPad Prism software). For the ARIH1 titration and CRY1 substrate, the fraction of $^{32}$P-labelled ubiquitin that had been converted to ubiquitylated CRY1 was quantified and then normalized to the value corresponding to the highest concentration of ARIH1. For all reactions, the final SCF and $^{32}$P-labelled substrate (or CRY1) concentrations were 0.25 μM, E1 was 0.5 μM, unlabelled ubiquitin was 30 μM, UBE2L3 was 5 μM, and cold competitor peptide was 100 μM. All experiments were performed in triplicate.

**Estimating the rate of ubiquitin transfer ($k_{obs}$) from ARIH1 and UBE2D3 to substrate.** Reactions that were single encounter between $^{32}$P-labelled substrate and SCF E3 ligase were assembled in reaction buffer as described in 'Estimating the $K_m$ of ARIH1 and UBE2D for various SCF–substrate complexes'. For cyclin E, ubiquitylated cyclin E and cyclin E (short) peptide substrates and ARIH1, E1 (1 μM), ubiquitin (60 μM) and UBE2L3 (10 μM) were incubated for 5 min (tube 2) at room temperature before the addition of cold competitor substrate (200 μM). In a separate tube (tube 1), neddylated CUL1–RBX1, SKP1–FBXW7(ΔD) and $^{32}$P-labelled substrate (0.5 μM) were incubated for 5 min followed by the addition of ARIH1 (5 μM). Each mixture was loaded into separate channels of a Kintek rapid-quench flow instrument (RQF-3) followed by the collection of time points that were quenched in 2× SDS–PAGE loading buffer. Substrate and products were resolved on 18% SDS–PAGE gels and quantified (Image Quant TL software; GE). The fraction of labelled peptide substrate that had been converted to product was then plotted as a function of time (Mathematica) and fit to a previously described closed-form solution to estimate the rate of substrate priming ($k_{obs}$)[58]. The same procedure was used for cyclin E, ubiquitylated cyclin E and cyclin E (short) peptide substrates and UBE2D3, except that UBE2D3 (20 μM) was used instead of UBE2L3 and ARIH1 was not added to tube 1. The procedure used for cyclin E was also used for β-catenin and β-catenin (short) peptide substrates and ARIH1 except that SKP1–β-TRCP2 was used and ARIH1 was added to tube 2 following the addition of cold competitor peptide. An identical procedure was used for β-catenin and β-catenin (short) peptide substrates and UBE2D3 as for cyclin E except that SKP1–β-TRCP2 was used and the final SCF ligase and labelled substrate concentrations were 0.5 μM and 0.1 μM, respectively. For CRY1, normalized ubiquitylated CRY1 was fit to a single-phase exponential decay function (GraphPad Prism software) considering CRY1 as if it was not dissociating from SCF ligase during the time course. All experiments were performed in triplicate.

#### Biochemical assays
**Ubiquitylation assays.** Ubiquitin transfer from UBE2L3 to ARIH1 was monitored using a pulse-chase format, in which ubiquitin is fluorescently labelled. First, the thioester-linked UBE2L3-ubiquitin intermediate is generated in the pulse reaction. This was carried out by incubating 10 μM UBE2L3, 0.3 μM UBA1 and 15 μM fluorescent ubiquitin in 25 mM HEPES pH 7.5, 100 mM NaCl, 2.5 mM MgCl$_2$, 1 mM ATP at room temperature for 15 min. The pulse reaction was quenched by incubation with 50 mM EDTA on ice for at least 5 min and subsequently diluted to 0.67 μM UBE2L3 in 25 mM MES pH 6.5, 100 mM NaCl. The chase initiation mix consisted of 1 μM NEDD8–CUL1-RBX1 and 0.75 μM ARIH1 (residue 90 to C terminus) in the same buffer but the quenched pulse reaction was further diluted to yield a final E2-ubiquitin concentration of 80 nM.

All reactions were carried out on ice. SDS–PAGE was performed under non-reducing conditions and the gel scanned on an Amersham Typhoon imager (GE Healthcare).

Examination of ubiquitin transfer to substrate is also initiated with the UBE2L3-Ub intermediate, because the ARIH1-ubiquitin intermediate is unstable[34]. However, substrate is included in the chase reaction to monitor this final ubiquitin transfer. The chase reaction mix contained 1 μM NEDD8–CUL1–RBX1, 1 μM substrate receptor (SKP1–FBXW7(ΔD)), 1 μM substrate (single Lys cyclin E phosphopeptide) and 0.75 μM ARIH1, in 25 mM HEPES pH 7.5, 100 mM NaCl. The chase reaction mix was equilibrated on ice for at least 20 min. The final concentrations were 0.4 μM UBE2L3, 0.4 μM NEDD8–CUL1-RBX1, 0.4 μM SKP1–FBXW7(ΔD), 0.4 μM cyclin E phosphopeptide substrate and 0.3 μM ARIH1. Reactions were either carried out on ice (Extended Data Fig. 6b, c) or at room temperature (Extended Data Fig. 6d–f) Reactions were quenched with 2× SDS–PAGE sample buffer at each dedicated time point.

**Assay for kinetic properties of the TS1 and TS2 ABP.** To compare the time courses of ABP and actual ubiquitin transfer reactions, the ABPs were tested with similar conditions as in the rapid-quench flow experiments. For TS1, 400 nM NEDD8–CUL1–RBX and 300 nM ARIH1 were incubated on ice for 15 min in 25 mM HEPES pH 7.5, 150 mM NaCl. After incubation, 400 nM (final concentration) of TS1 ABP was added to the reaction and quenched at indicated time point with SDS–PAGE sample buffer. For TS2, 400 nM NEDD8–CUL1–RBX, 400 nM SKP1–FBXW7(ΔD) and 300 nM ARIH1 were incubated on ice for 15 min in 25 mM HEPES pH 7.5, 150 mM NaCl. After incubation, 2 μM (final concentration) of TS2 ABP was added to the reaction and quenched at indicated time point with SDS–PAGE sample buffer.

**Assays for catalytic configuration for TS1, probed by reactivity with TS1 ABP.** In the first transition state, ubiquitin is transferred from UBE2L3 to ARIH1. The native configuration was validated by testing requirement for essential elements of the TS1 reaction. Reaction components (ARIH1 (either full-length wild type, full-length ARHI1(C357S) or other mutants) and NEDD8–CUL1–RBX1) were stoichiometrically mixed (1 μM per component) in 25 mM HEPES pH 7.8, 150 mM NaCl and preincubated on ice for at least 10 min. After incubation, 5–10-fold excess of TS1 ABP was added to the reaction mix to start the reaction. Reactions were carried out at room temperature for 1 h. Quenching was performed by the addition of 2× SDS–PAGE sample buffer.

**ABP assays probing features of TS2.** In the second transition state, ubiquitin is transferred from ARIH1 to an SCF-bound substrate. Because ARIH1 can transfer ubiquitin to numerous substrates recruited to diverse SCF ligases, several TS2 substrate ABPs were synthesized and tested for their requirements to react with ARIH1 and their specificity to react only in presence of the cognate SKP1–F-box-protein substrate receptor (Extended Data Fig. 2e, h). The reaction components (ARIH1(residue 90 to C terminus), either NEDD8–CUL1–RBX1 or CUL1–RBX1 and the SKP1–F-box-protein substrate receptor) were stoichiometrically mixed (2 μM per component) in 25 mM HEPES pH 7.8, 150 mM NaCl, 1 mM TCEP and preincubated on ice for at least 10 min. After incubation, equimolar amounts of TS2 ABPs were added to the reaction mix in Extended Data Fig. 6f and twofold excess of TS2 ABP in Extended Data Fig. 2g, h. The reaction was quenched by the addition of 2× SDS–PAGE sample buffer at dedicated time points. All reactions were carried out at room temperature. Additionally, after validation that TS2 ABPs capture the native configuration, the reaction with this ABP serves as an assay for native configuration, which cannot otherwise be explicitly tested by enzymology owing to the two-step nature of the reaction. Thus, reactivity with TS2 ABP can probe the TS2 configuration for the regions of the proteins that contribute to both transition states (Extended Data Fig. 6f).

## Cryo-EM

**Preparation of samples capturing pre-TS1.** Following the enzymatic cascade of ubiquitylation reactions, the E1 enzyme links ubiquitin to the E2 enzyme in an ATP-dependent manner. Here we mimicked this reactive E2-ubiquitin intermediate with ubiquitin linked by a stable isopeptide bond to UBE2L3(C86K), which was prepared as previously described[49]. After pre-equilibration of ARIH1–NEDD8–CUL1–RBX1–SKP1–FBXW7(ΔD)–cyclin E phosphopeptide at equimolar ratio (10 μM final component concentration) in sizing buffer (25 mM HEPES pH 7.8, 150 mM NaCl, 1mM TCEP) for 10 min on ice, isopeptide-linked UBE2L3-ubiquitin was added in a final concentration of 15 μM. The sample was purified via size-exclusion chromatography, concentrated and subjected for crosslinking via GraFix[59]. GraFix peak fractions were desalted (Zeba Spin Columns, 0.5 ml, 7-kDa molecular weight cut-off, Thermo) into size-exclusion buffer and concentrated to 0.4 mg ml⁻¹. Three μl of sample was applied to R1.2/1.3 holey carbon grids (Quantifoil), blotted for 3 s at about 100% humidity and 4 °C, and plunge-frozen in liquid ethane by using a Vitrobot Mark IV.

**Preparation of samples capturing TS1 (neddylated SCF-catalysed ubiquitin transfer from UBE2L3 to ARIH1).** Samples with two structurally distinct F-box protein–substrate assemblies were prepared. The structure in Fig. 1b shows neddylated SCF[FBXW7]-activated ubiquitin transfer from UBE2L3 to ARIH1. Here, the 'F-box protein–substrate' assembly represents a monomeric version of SKP1–FBXW7–cyclin E phospho-degron. In Fig. 2, the F-box protein–substrate assembly is SKP1–SKP2–CKSHS1–cyclin A–CDK2–phopsho-p27.

TS1 ABP–UBE2L3-ubiquitin complex samples were generated by pre-equilibrating subcomplexes (either ARIH1–NEDD8–CUL1–RBX1–SKP1–SKP2–CKSHS1(5–73)–p27(22–106) N-terminal domain and p27 phoshopeptide–CDK2–cyclin A(residue 170 to C terminus) or ARIH1–NEDD8–CUL1–RBX1–SKP1–FBXW7(ΔD)–cyclin E phosphopeptide) at equimolar ratio (10 μM final component concentration) in 25 mM HEPES pH 7.8, 150 mM NaCl, for 10 min on ice followed by the addition of tenfold molar excess TS1 ABP to initiate three-way cross-linking. To ensure maximal conversion of ARIH1, samples were incubated for 2 h at room temperature. Samples were then purified via size-exclusion chromatography on a Superose 6 Increase column and crosslinked via GraFix. GraFix peak fractions were desalted and sample concentrated to 0.4 mg ml⁻¹. Sample was plunged as described for pre-TS1.

**Preparation of samples capturing post-TS1.** After TS1, ubiquitin is conjugated to the catalytic cysteine of ARIH1. This intermediate was mimicked by using the ABP ubiquitin–VME, that can modify the catalytic cysteine of ARIH1 in a neddylated SCF-dependent manner[3,4]. After pre-equilibration of ARIH1–NEDD8–CUL1–RBX1–SKP1–FBXW7(ΔD)–cyclin E phosphopeptide at equimolar ratio (10 μM final component concentration) in sizing buffer (25 mM HEPES pH 7.8, 150 mM NaCl, 1mM TCEP) for 10 min on ice, tenfold molar excess of ubiquitin–VME was added to the other components. The reaction was incubated for 2 h at room temperature to fully convert ARIH1 to ARIH1-ubiquitin–VME. Consequent sample treatments were the same as for the TS1 ABP samples.

**Preparation of samples capturing TS2 (ARIH1-mediated ubiquitylation of SCF substrates).** Samples with two structurally distinct F-box protein assemblies, and with a neddylated SCF ligase or a mutant of ARIH1 that bypasses the requirement for neddylation in TS1 to allow for TS2 (ref. [4]) (ARIH1(F430A/E431A/E503A)) were prepared. TS2 ABP complex samples were prepared by pre-equilibrating subcomplexes on ice (either ARIH1(residue 92 to C terminus)–NEDD8–CUL1–RBX1–SKP1–FBXW7(ΔD), ARIH1(F430A/ E431A/ E503A, residue 92 to C terminus)–CUL1–RBX1–SKP1–FBXW7(ΔD) or ARIH1(residue 92 to C terminus*)–CUL1–RBX1–SKP1–SKP2(residue 101 to C terminus)–CKSHS1(5–73)). This was carried out for 10 min on ice at equimolar

ratio (10 μM components) in 25 mM HEPES pH 7.8, 150 mM NaCl, 1 mM TCEP. After addition of the matching TS2 ABP in fivefold molar excess, maximal conversion of ARIH1 was achieved by incubation for 2.5 h on ice. Subsequent treatment of TS2 ABP samples was identical to TS1 ABP samples with the exception of concentrating crosslinked, desalted complexes to only 0.3 mg ml⁻¹ before plunging.

It was crucial for all ABP complex samples to convert as much ARIH1 to the stable transition state mimic as possible before complex purification. This was done by optimizing reaction time, buffer and temperature. If unmodified ARIH1 was present after an ABP reaction, it was essential to purify it away from the transition state mimic to obtain a homogeneous sample for cryo-EM.

**Data collection.** Screening datasets were collected on a Talos Arctica or Glacios transmission electron microscope (TEM) at 200 kV equipped with a Falcon II direct detector (linear mode) or Gatan K2 (counting mode). Approximately 1,000 micrographs were collected per dataset with a pixel size of 1.997 Å, defocus range of −1.5 to −3.5 μm and a total exposure of around 60–70 e⁻ Å⁻² split across 40 frames. After screening, larger datasets were collected on a Titan/Krios TEM at 300 kV equipped with a post-GIF Gatan K3 Summit direct electron detector (counting mode). Between 9,000 and 11,000 movies were collected per sample with either 0.851 Å or 1.09 Å pixel size, a total exposure ranging from 60 to 80 e⁻ Å⁻² and defocus values from −0.8 to −3.2 μm.

**Data processing.** RELION 3.00[60] was used to align and dose-weight raw movie frames. Each drift-corrected micrograph was then contrast-transfer-function-corrected via Gctf[61]. Particle picking was performed with Gautomatch (K. Zhang). RELION 3.0 was ultimately used to do 2D classification, initial model building, 3D classification, global and local 3D refinement, particle polishing and post-processing.

**Model building.** To extract the features of the TS1 ABP–ARIH1 neddylated SCF[SKP2] complex at high resolution, three initial models were built into focused refined maps. A CUL1 N-terminal domain (residues 17–339)–SKP1–SKP2–CKSHS1–phospho-p27–cyclin A–CDK2 portion of the complex was built using a map at 3.6 Å resolution. Another model comprising all CUL1 except the neddylated WHB domain (residues 17–686), all domains from RBX1 (residues 21–108) and the IBR (residues 98–106, 271–328) and Ariadne domains (residues 406–554) of ARIH1 was built using a map at 3.6 Å resolution. A third model containing the C-terminal portions of CUL1–RBX1 (part of CR3 subdomain of CUL1, the 4HB subdomain of CUL1, the C/R domain and the NEDD8-linked WHB domain corresponding to residues 336–686 and 705–776 and RBX1 residues 21–108), its isopeptide-bonded NEDD8 and ARIH1 capturing UBE2L3-linked ubiquitin was built using a map at 3.6 Å resolution, which was subjected to further focused refinement focused over the active site.

Initial models were generated by using structures of previous subcomplexes and components (PDB 1LDJ, 5UDH, 2AST, 6TTU and 1H27). Structures were manually placed into the maps and then fit by rigid-body refinement using UCSF Chimera[62]. Real-space refinements and further manual model building was performed for accurate geometry and map-to-model correlation. COOT70[63] was used for manual modelling and Phenix.refine71[64] was used for real space refinement. Invisible loops were excluded from models, but side chains were included both where clearly resolved in the maps, and where backbone was visible they were placed on the basis of the previous crystal structures.

A composite map was generated by merging several focused maps (Extended Data Table 1, Supplementary Fig. 2) via the PHENIX combine focused maps feature[64]. The final refined map of the class, showing prominent density for the catalytic core, was used as a base map onto which the focused maps were resampled. This composite map displayed main and side chain densities for most part, which enabled placing the

subcomplexes that had been built and fully refined using the focused maps. Ultimately, the composite map was used for final refinement of atomic coordinates for the full TS1 ABP–ARIH1 neddylated SCF[SKP2] complex. The final refined model for the full TS1 ABP–ARIH1 neddylated SCF[SKP2] complex also matches the consensus map.

The model for the TS2 p27 ABP SCF[SKP2] complex was built and refined into the consensus map at 3.9 Å resolution. The structures of components from the TS1 intermediate were manually placed into the maps and then fit by rigid-body refinement using UCSF Chimera[62]. In terms of the catalytic elements, map quality allowed real-space refinement of the E3–E3act super-domain but only docking of the ubiquitin transferase domain (Extended Data Fig. 3f, Supplementary Fig. 3).

The ribbon models in Fig. 1—showing intermediates along ARIH1 ubiquitylation of a phosphopeptide substrate of SCF[FBXW7]—were created by fitting refined coordinates into the maps corresponding to the indicated intermediates. The cyclin E phosphopepeptide-bound substrate-binding domain was modelled by fitting a previous crystal structure of SKP1–FBXW7–cyclin E (PDB 2OVQ[47]) into eletron micros-copy density. The remaining portions of the models derived from the fully refined coordinates for the TS1 and TS2 intermediates for complexes with SCF[SKP2]. The models are shown in low-pass-filtered cryo-EM maps in Extended Data Fig. 4.

### Reporting summary

Further information on research design is available in the Nature Research Reporting Summary linked to this paper.

### Data availability

The atomic coordinates and electron microscopy maps have been deposited in the PDB with accession codes 7B5N, 7B5R, 7B5S (sub-complexes of the TS1 model), 7B5L (TS1) and 7B5M (TS2) and in the Electron Microscopy Data Bank with codes EMD-12004 (pre-TS1), EMD-12037/12038/12041/12048/12050 (composite map/consensus map/three focused maps/TS1 neddylated SCF[SKP2] with p27 phospho-peptide substrate/wild-type ARIH1), EMDB-12036 (TS1 neddylated SCF[FBXW7] with cyclin E phosphopeptide substrate and wild-type ARIH1), EMDB-12005 (post-TS1), EMDB-12040 (TS2, SCF[SKP2] with p27 phospho-peptide substrate and ARIH1(F430A/E431A/E503A) residue 92 to C terminus), EMDB-12006 (TS2, SCF[FBXW7] with cyclin E phosphopeptide substrate and ARIH1(F430A/E431A/E503A) residue 92 to C terminus) and EMDB-12039 (TS2, neddylated SKP1–CUL1–FBXW7(ΔD) with cyc-lin E phosphopeptide substrate and ARIH1 residue 92 to C terminus). Uncropped gel source data are included as Supplementary Informa-tion. All other reagents and data (for example, raw gels of replicate experiments and raw movie electron microscopy data) are available from the corresponding author upon request.

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

**Acknowledgements** This study is dedicated to the memory of Huib Ovaa, an exceptional and inspiring scientist, colleague, mentor and friend. We thank J. Kellermann, H. Stark, A. Chari, S. Kostrhon, S. von Gronau, J. Liwocha, L. Neumaier, S. Uebel, S. Pettera, C. M. P. Talavera Ormeno, P. J. M. Hekking, V. Sanchez and N. Purser for assistance and reagents; J. W. Harper and R. V. Farese Jr for helpful discussions and critical reading of the manuscript; and D. Bollschweiler and T. Schäfer of the cryo-EM facility at Max Planck Institute of Biochemistry. This study was supported by the Max Planck Gesellschaft, the ERC (H2020 789016-NEDD8Activate) and Leibniz Prize from the DFG (SCHU 3196/1-1) to B.A.S., and NIH R15GM117555-02 to G.K. D.C.S. was supported by ALSAC/St Jude.

**Author contributions** D.T.K. and B.A.S. conceived the project. D.H.-G., D.T.K., K.B., M.K., D.C.S. and G.K. performed protein biochemistry. D.T.K. and M.P.C.M. designed and D.T.K., M.P.C.M. and D.H.-G. generated ABPs for stable transition-state mimics, supervised by H.O. and B.A.S. D.H.-G., D.C.S. and G.K. performed ubiquitylation and ABP assays. D.H.-G. collected, processed and refined cryo-EM data, and J.R.P. built and refined structures. D.H.-G., G.K. and B.A.S. analysed data and prepared the manuscript with input from all authors. B.A.S. supervised the project.

**Funding** Open access funding provided by Max Planck Society.

**Competing interests** H.O. was a shareholder of UbiqBio. All other authors declare no competing interests.

**Additional information**
**Correspondence and requests for materials** should be addressed to B.A.S.

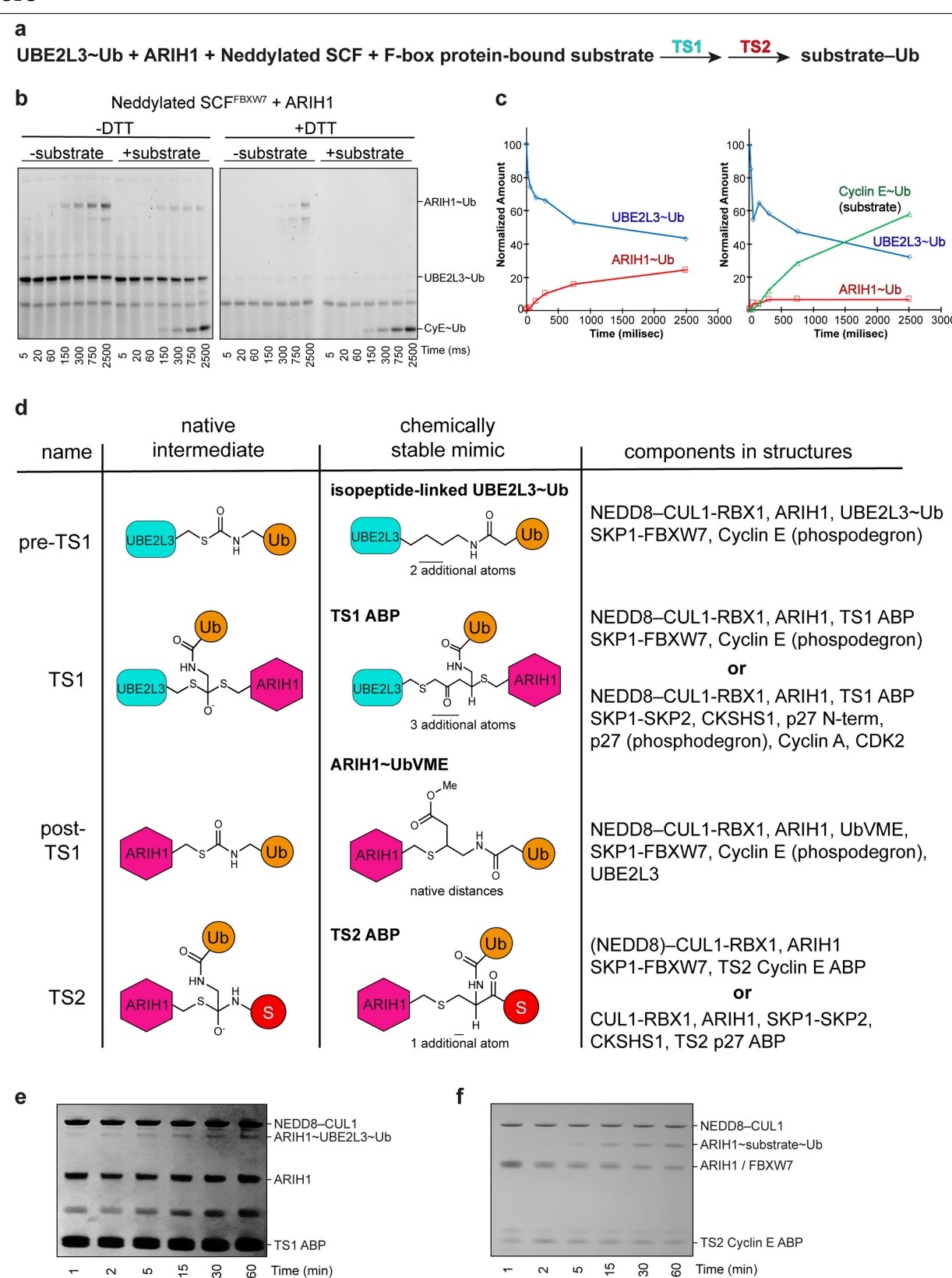

**a**

UBE2L3~Ub + ARIH1 + Neddylated SCF + F-box protein-bound substrate $\xrightarrow{\text{TS1}}\xrightarrow{\text{TS2}}$ substrate–Ub

**b**

Neddylated SCF^FBXW7 + ARIH1

**c**

**d**

| name | native intermediate | chemically stable mimic | components in structures |
|---|---|---|---|
| pre-TS1 | | **isopeptide-linked UBE2L3~Ub** 2 additional atoms | NEDD8–CUL1-RBX1, ARIH1, UBE2L3~Ub SKP1-FBXW7, Cyclin E (phospodegron) |
| TS1 | | **TS1 ABP** 3 additional atoms | NEDD8–CUL1-RBX1, ARIH1, TS1 ABP SKP1-FBXW7, Cyclin E (phospodegron) **or** NEDD8–CUL1-RBX1, ARIH1, TS1 ABP SKP1-SKP2, CKSHS1, p27 N-term, p27 (phosphodegron), Cyclin A, CDK2 |
| post-TS1 | | **ARIH1~UbVME** native distances | NEDD8–CUL1-RBX1, ARIH1, UbVME, SKP1-FBXW7, Cyclin E (phospodegron), UBE2L3 |
| TS2 | | **TS2 ABP** 1 additional atom | (NEDD8)–CUL1-RBX1, ARIH1 SKP1-FBXW7, TS2 Cyclin E ABP **or** CUL1-RBX1, ARIH1, SKP1-SKP2, CKSHS1, TS2 p27 ABP |

**e**

**f**

**Extended Data Fig. 1** | See next page for caption.

**Extended Data Fig. 1 | Stable mimics for fleeting E3–E3 ubiquitylation intermediates and transition states (TS1 and TS2). a**, Neddylated SCF–ARIH1 RBR E3–E3-catalysed ubiquitylation of an F-box-protein-bound substrate proceeds through two transition states. First, the C terminus of ubiquitin is transferred from the catalytic cysteine of the E2 UBE2L3 to the catalytic cysteine of ARIH1 (TS1), and then from ARIH1 to substrate (TS2). **b**, SDS–PAGE gels (left, nonreducing; right, reducing) of rapid-quench flow experiments resolving intermediates in fluorescent ubiquitin transfer from UBE2L3 to ARIH1 to cyclin E phosphopeptide substrate of neddylated SCF$^{FBXW7}$ (CUL1–SKP1–FBXW7(ΔD)) (millisecond time scale). Gel images are representative of independent biological replicates ($n = 2$). **c**, Quantification of results obtained from rapid-quench flow and monitoring fluorescent ubiquitin transfer, with or without cyclin E substrate included in the reaction. **d**, Chemical structures of native ubiquitylation intermediates and stable mimics used in this study. In the pre-TS1 intermediate, the C terminus of ubiquitin is linked to the catalytic cysteine of UBE2L3 by a thioester bond; in the stable mimic, the C terminus of ubiquitin is linked by an isopeptide bond to a lysine replacement for the catalytic cysteine of UBE2L3. The TS1 intermediate is mimicked by using an ABP with an electrophilic moiety installed between the C terminus of ubiquitin and the catalytic cysteine of UBE2L3 to trap the catalytic cysteine of ARIH1 via a stable three-way cross-link. Following the native TS1 reaction, in the post-TS1 intermediate, ubiquitin is thioester-bonded to the catalytic cysteine of ARIH1. The stable mimic used ubiquitin–VME to stably couple ubiquitin to the catalytic cysteine of ARIH1. In the TS2 reaction, ubiquitin is transferred from ARIH1 to the substrate. Our electrophilic TS2 ABP forms a stable mimic by simultaneously linking the catalytic cysteine of ARIH1, the C terminus of ubiquitin and the acceptor site on a peptide substrate. **e**, SDS–PAGE gel monitoring the formation of the stable TS1 mimic, in which ARIH1, ubiquitin and UBE2L3 are linked via a single atom. For comparison, reactions were carried out at the same concentrations of ARIH1 and neddylated CUL1–RBX1, and similar buffer and temperature conditions as in **b**. Gel image is representative of independent technical replicates ($n = 2$). Higher protein concentrations and extended times were used to generate samples for cryo EM. **f**, SDS–PAGE gel tracking neddylated SCF-dependent generation of the stable TS2 mimic, in which the catalytic cysteine of ARIH1, the C terminus of ubiquitin and the peptide substrate are linked. Gel image is representative of independent technical replicates ($n = 2$).

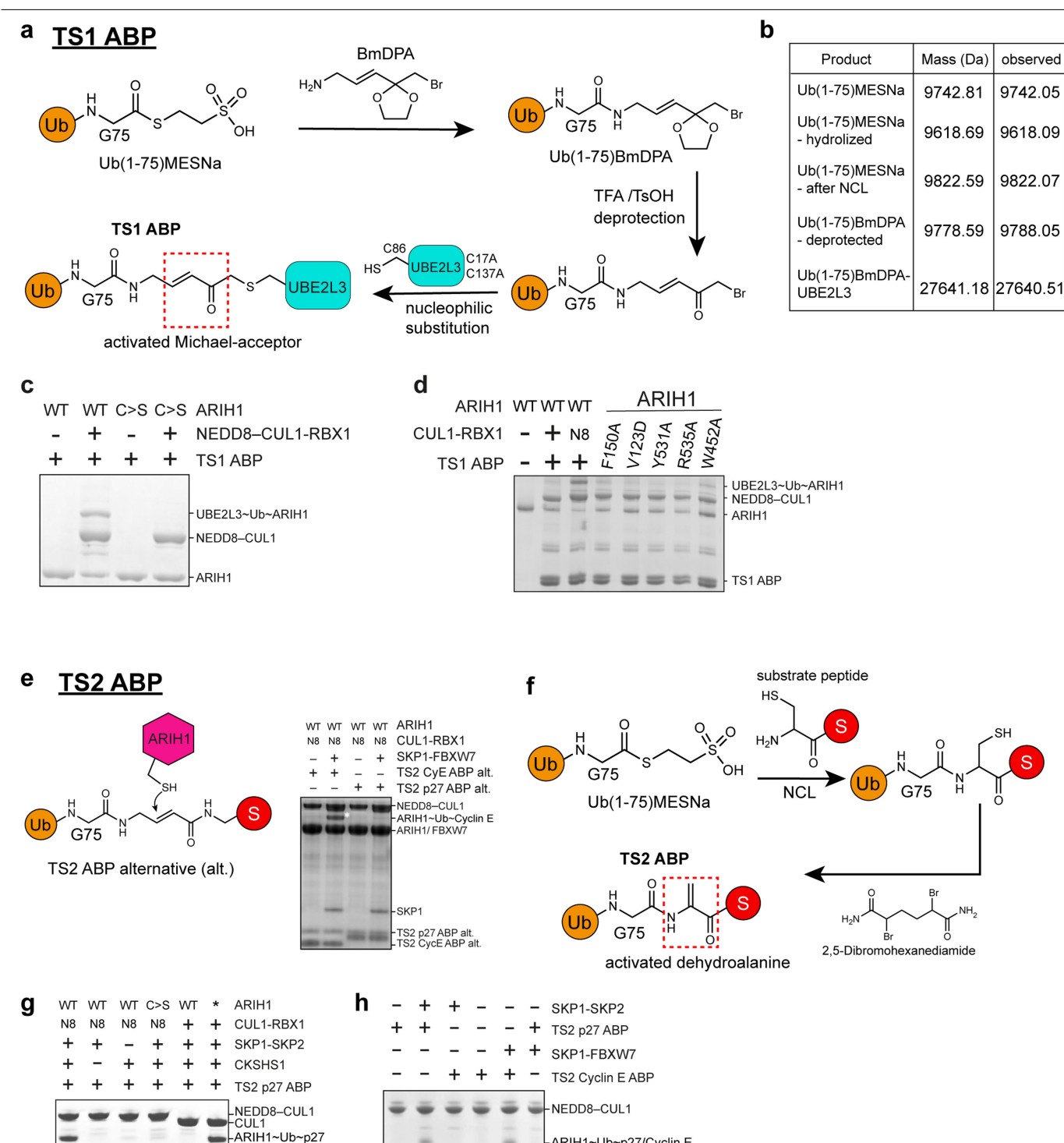

**a  TS1 ABP**

BmDPA

Ub(1-75)MESNa → Ub(1-75)BmDPA

TFA /TsOH deprotection

**TS1 ABP**

activated Michael-acceptor

nucleophilic substitution

**b**

| Product | Mass (Da) | observed |
|---|---|---|
| Ub(1-75)MESNa | 9742.81 | 9742.05 |
| Ub(1-75)MESNa - hydrolized | 9618.69 | 9618.09 |
| Ub(1-75)MESNa - after NCL | 9822.59 | 9822.07 |
| Ub(1-75)BmDPA - deprotected | 9778.59 | 9788.05 |
| Ub(1-75)BmDPA-UBE2L3 | 27641.18 | 27640.51 |

**c**

| WT | WT | C>S | C>S | ARIH1 |
|---|---|---|---|---|
| – | + | – | + | NEDD8–CUL1-RBX1 |
| + | + | + | + | TS1 ABP |

UBE2L3~Ub~ARIH1
NEDD8–CUL1
ARIH1

**d**

ARIH1 WT WT WT / ARIH1
CUL1-RBX1 – + N8
TS1 ABP – + + F150A V123D Y531A R535A W452A

UBE2L3~Ub~ARIH1
NEDD8–CUL1
ARIH1
TS1 ABP

**e  TS2 ABP**

ARIH1

TS2 ABP alternative (alt.)

ARIH1
WT WT WT WT
N8 N8 N8 N8 CUL1-RBX1
– + – + SKP1-FBXW7
+ + – – TS2 CyE ABP alt.
– – + + TS2 p27 ABP alt.

NEDD8–CUL1
ARIH1~Ub~Cyclin E
ARIH1/ FBXW7
SKP1
TS2 p27 ABP alt.
TS2 CycE ABP alt.

**f**

substrate peptide

Ub(1-75)MESNa → NCL

2,5-Dibromohexanediamide

**TS2 ABP**

activated dehydroalanine

**g**

| WT | WT | WT | C>S | WT | * | ARIH1 |
|---|---|---|---|---|---|---|
| N8 | N8 | N8 | N8 | + | + | CUL1-RBX1 |
| + | + | – | + | + | + | SKP1-SKP2 |
| + | – | + | + | + | + | CKSHS1 |
| + | + | + | + | + | + | TS2 p27 ABP |

NEDD8–CUL1
CUL1
ARIH1~Ub~p27
ARIH1
SKP2

**h**

| – | + | + | – | – | – | SKP1-SKP2 |
|---|---|---|---|---|---|---|
| + | + | – | – | – | + | TS2 p27 ABP |
| – | – | – | – | + | + | SKP1-FBXW7 |
| – | – | + | + | + | – | TS2 Cyclin E ABP |

NEDD8–CUL1
ARIH1~Ub~p27/Cyclin E
ARIH1
substrate receptor

**Extended Data Fig. 2** | See next page for caption.

**Extended Data Fig. 2 | Synthesis of ABPs to capture TS1 and TS2. a**, Strategy to generate an ABP to visualize TS1 (TS1 ABP). The goal was to generate an ABP with a warhead between the catalytic cysteine of UBE2L3 and the C terminus of ubiquitin that would react with ARIH1 only when assembled with a neddylated CRL. An intein-based semisynthesis route was used to couple Ub(1–75)–MESNa and (E)-3-[2-(bromomethyl)-1,3-dioxolan-2-yl]prop-2-en-1-amine (BmDPA) to yield a cyclic ketal-protected ubiquitin species. Acidic deprotection of the cyclic ketal yields a reactive ubiquitin species[65], which when conjugated to a single-cysteine-containing version of UBE2L3 produces an ABP with a Michael acceptor between the C terminus of ubiquitin and the active site of UBE2L3. **b**, Quality controls comparing predicted masses for Ub–MESNa, ABP precursor and TS1 ABP entities with measurements obtained by electrospray ionization–time-of-flight mass spectrometry. **c**, SDS–PAGE gel confirming TS1 ABP reaction depends on the catalytic cysteine of ARIH1 (C>S refers to serine replacement). Gel image is representative of independent technical replicates (n = 2). **d**, SDS–PAGE gel demonstrating TS1 ABP reaction with ARIH1 depends on neddylated CUL1–RBX1 and ARIH1 residues required for ubiquitylating client substrates bound to an F-box protein[4]. Gel image is representative of independent technical replicates (n = 2). **e**, Because structural biology is an empirical endeavour, various TS2 ABP approaches were tested to identify a strategy yielding high-quality electron microscopy data visualizing TS2. The concept was to place a warhead between a substrate and the C terminus of ubiquitin, to generate an ABP that would react only with ARIH1 super-assembled with the SCF containing the cognate F-box protein of the substrate. The fully synthetic TS2 ABP alternative (left) displayed reactivity and specificity matching the native reaction when assembled with cyclin E or p27 phosphopeptide substrate mimics, as shown by SDS–PAGE gel (right). **f**, In parallel, we tested a semisynthetic strategy, which led to high-resolution cryo-EM structures, and thus complexes generated with this strategy are referred to as TS2 throughout the article. Ub–MESNa and a substrate phosphopeptide with an N-terminal cysteine placed to mimic the acceptor site were fused via native chemical ligation and the free cysteine was converted to dehydroalanine. **g**, SDS–PAGE showing TS2 p27 ABP reaction with ARIH1 requires all elements needed for native TS2, or use of a mutant version of ARIH1 (ARIH1(F430A/E431A/E503A)) bypassing the need for NEDD8 for this reaction. Gel image is representative of independent technical replicates (n = 2). **h**, SDS–PAGE testing specificity of TS2 ABPs for cognate F-box proteins. Phosphorylated cyclin E and p27 are substrates of SCF[FBXW7] and SCF[SKP2], respectively. All experiments with SCF[SKP2] also contained the essential protein partner CKSHS1 unless otherwise indicated. Gel image is representative of independent technical replicates (n = 2).

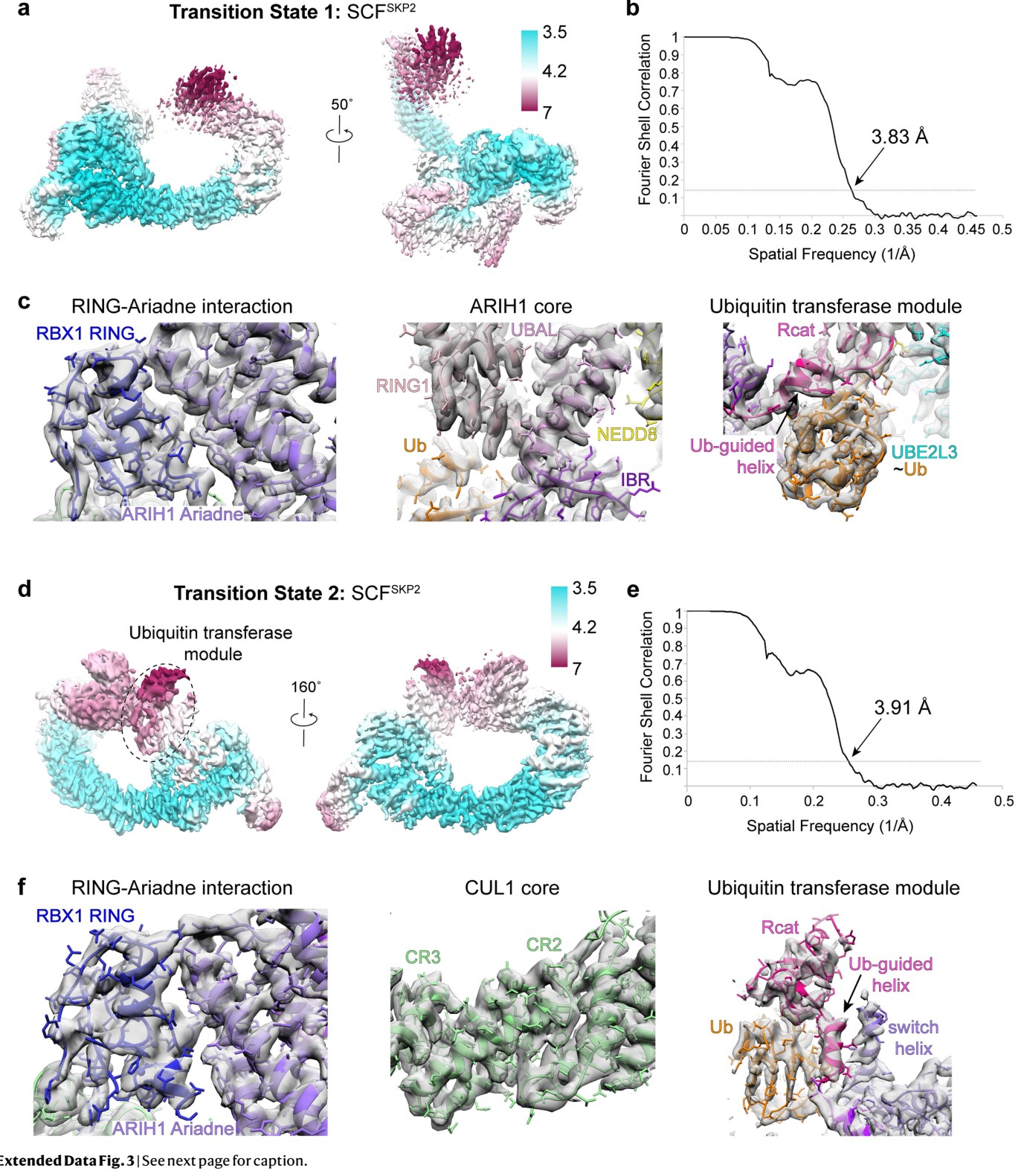

**a** Transition State 1: SCF^SKP2

**b**

**c** RING-Ariadne interaction ARIH1 core Ubiquitin transferase module

**d** Transition State 2: SCF^SKP2

**e**

**f** RING-Ariadne interaction CUL1 core Ubiquitin transferase module

**Extended Data Fig. 3** | See next page for caption.

**Extended Data Fig. 3 | Cryo-EM map quality analysis. a**, Cryo-EM map representing TS1, coloured by local resolution in Å as estimated by ResMap. The map was generated using our TS1 ABP for the complex representing UBE2L3-ubiquitin-ARIH1 bound to neddylated CUL1–RBX1–SKP1–SKP2–CKSHS1–p27–cyclin A–CDK2. **b**, Fourier shell correlation curve (FSC) displaying an overall resolution of 3.83 Å with the FSC = 0.143 criterion. **c**, Structure shown in electron microscopy density. Left, close-up of E3–E3act domain, showing side-chain density for interactions between the RING domain of RBX1 and the Ariadne domain of ARIH1. Middle, close-up of E3–E3 platform showing interactions with UBE2L3-linked ubiquitin. Right, ubiquitin transferase module. The map quality permitted modelling of side chains either visible in the density or by wholesale docking of previous crystal structures. **d**, Cryo-EM map representing TS2, coloured by local resolution in Å as estimated by ResMap. The map was generated using our TS2 p27 ABP for the complex representing ARIH1-ubiquitin-p27 bound to CUL1–RBX1–SKP1–SKP2-CKSHS1. This particular map is from a complex using ARIH1(F430A/E431A/E503A), which was previously shown to relieve autoinhibition and bypass the need for neddylation[4]. **e**, FSC curve displaying an overall resolution of 3.91 Å with the FSC = 0.143 criterion. **f**, Structure shown in electron microscopy density, representing highest, medium and lowest resolution areas of the map. Left, close-up of E3–E3act domain, showing side-chain density for interactions between the RING domain of RBX1 and the Ariadne domain of ARIH1. Middle, portion of CUL1. Right, ubiquitin transferase module. After the ubiquitin transferase module was modelled and refined for the structure representing TS1, it was wholesale-docked into the lower-resolution density for this region in the cryo-EM maps representing TS2.

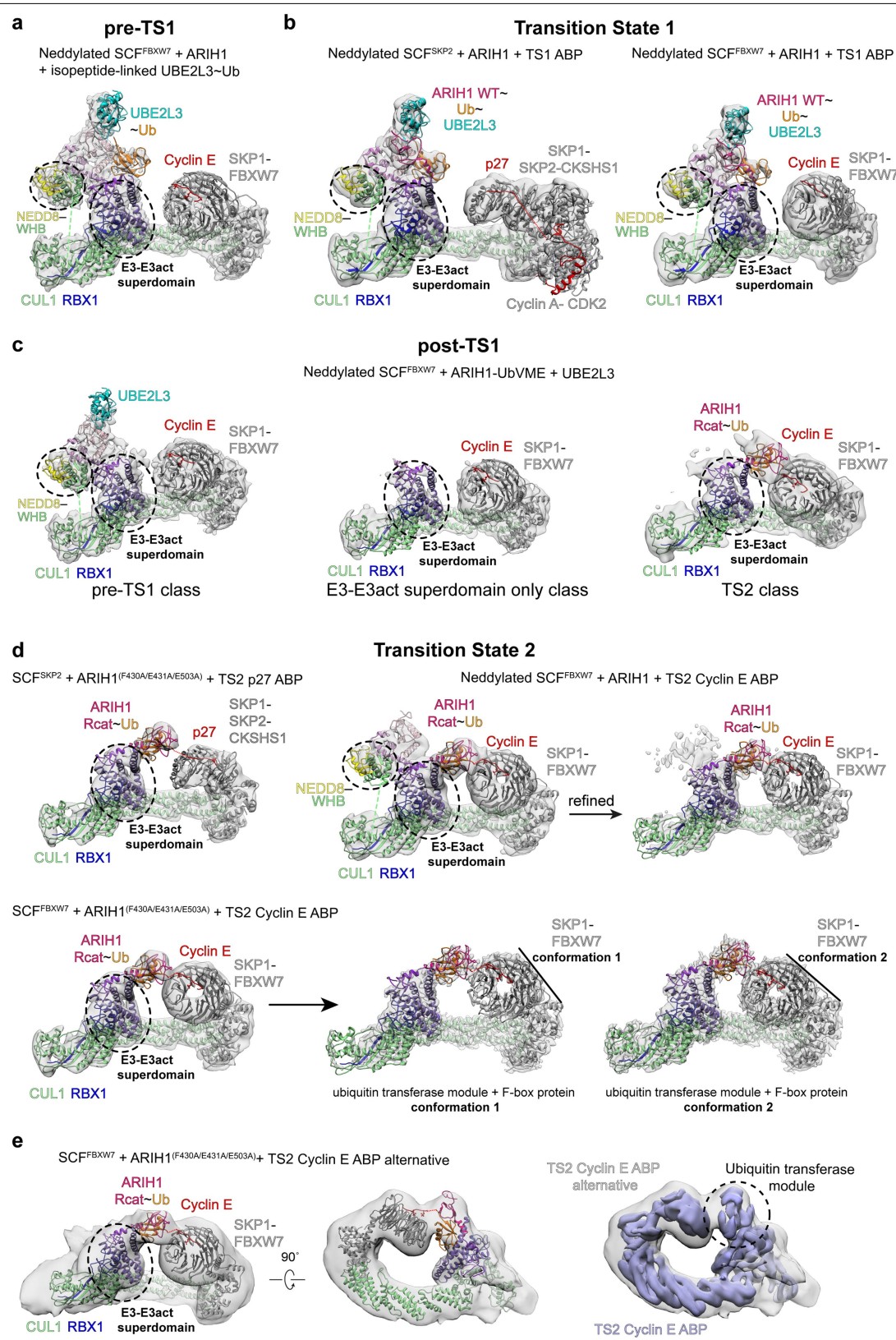

**Extended Data Fig. 4** | See next page for caption.

**Extended Data Fig. 4 | Cryo-EM maps representing intermediates in neddylated SCF-dependent ubiquitin transfer from UBE2L3 to ARIH1 and from ARIH1 to substrates recruited by structurally diverse F-box proteins. a**, Low-pass-filtered cryo-EM map representing intermediate with ubiquitin linked to E2 UBE2L3, before transfer to ARIH1 bound to neddylated SCF (pre-TS1). Model was generated by docking the following structures into the higher-resolution (4.5 Å) map: SKP1–FBXW7–cyclin E[47] (PDB 2OVQ), and UBE2L3-ubiquitin bound to ARIH1-neddylated CUL1–RBX1 refined from the structure representing TS1. The Ub-guided helix and Rcat domain of ARIH1 are not visible in this intermediate and were removed before fitting. **b**, Low-pass-filtered cryo-EM maps representing TS1 intermediate, neddylated SCF-dependent ubiquitin transfer from UBE2L3 to ARIH1, for SCF ligases with two structurally divergent F-box proteins. Left, ribbon diagram for the final refined model for one of these SCF[SKP2]. The concave leucine-rich repeat domain of the F-box protein SKP2 enwraps its partner CKSHS1 to corecruit the intrinsically disordered C-terminal domain of phosphorylated p27[48]. This assembly is sufficient for efficient phospho-p27 ubiquitylation in vitro[66], but the complex can be further augmented by additional protein–protein interactions. CKSHS1 can also bind cyclin A–CDK2, which also binds the N-terminal domain of p27[48,67,68]. This TS1 structure comprises a p27 N-terminal domain-bound cyclin A–CDK2, phospho-p27 C-terminal-domain-bound CKSHS1–SKP2–SKP1 bound to a neddylated CUL1–RBX1-activated ARIH1-ubiquitin-UBE2L3 assembly generated with the TS1 ABP. The right panel shows a comparable assembly with the monomeric (ΔD) version of the F-box protein FBXW7, in which the top side of the eight-bladed WD40 β-propeller domain of FBXW7 recruits a phosphopeptide substrate derived from cyclin E[47]. The model was made by docking SKP1–FBXW7–cyclin E (PDB 2OVQ) and catalytic portions of the final refined model for neddylated CUL1–RBX1-activated ARIH1-ubiquitin-UBE2L3 TS1 complex with SKP2 on the left. **c**, Cryo-EM data for the post-TS1 intermediate, with ubiquitin linked to the catalytic cysteine of ARIH1, yielded three classes. (1) In the most abundant 'E3–E3act super-domain only' class (middle), only the E3–E3act super-domain and SCF scaffold are visible. (2) A 'pre-TS1 class' largely resembles the pre-TS1 assembly, as the E3–E3act super-domain and E3–E3 platform as well as the E2 UBE2L3 are visible. The ubiquitin transferase module is not visible. (3) A low-resolution 'TS2 class' resembles the structural architecture of TS2, with density presumably corresponding to the

ubiquitin transferase module (the ubiquitin-guided helix and Rcat domain of ARIH1 bound to ubiquitin) poised adjacent to the F-box-protein-bound substrate. **d**, Low-pass-filtered cryo-EM maps for some TS2 complexes, representing ubiquitin transfer from ARIH1 to an SCF-bound substrate. Top left, ribbon diagram for the final refined model for one of these, with SKP1–CUL1–SKP2. This structure comprises ARIH1(F430A/E431A/E503A)-ubiquitin-phospho p27 peptide (generated with a TS2 ABP) bound to CKSHS1–SKP2–SKP1–CUL1–RBX1. ARIH1(F430A/E431A/E503A) has previously been shown to relieve autoinhibition and bypass the need for neddylation[4]. Top middle and top right, cryo-EM maps from the complex representing neddylated SCF[FBXW7]-dependent ubiquitin transfer from ARIH1 to cyclin E phosphopeptide substrate (captured with our cyclin-E-based TS2 cyclin E ABP). The ribbon diagram corresponds to SKP1–FBXW7(ΔD)–cyclin E from a previous crystal structure[47], CUL1–RBX1–ARIH1-ubiquitin from the refined structure with SKP2 (top left), and for the unrefined class, the neddylated CUL1–ARIH1 platform (the UBAL, RING1, RTI helix and IBR of ARIH1 and NEDD8 isopeptide-linked to the WHB domain of CUL1) from the refined structure of TS1 (top right). The latter portion of the complex is not visible upon refinement to higher resolution (top right). Bottom left, low-pass-filtered cryo-EM map and docked structures for SCF[FBXW7] and ARIH1(F430A/E431A/E503A) captured with our cyclin-E-based TS2 ABP, which superimposes with the refined class from neddylated SCF[FBXW7] and wild-type ARIH1, thus validating the use of ARIH1(F430A/E431A/E503A) to improve cryo-EM map quality for TS2. Middle, right, two conformations of TS2 obtained by major classification of the cryo-EM data are shown. Although the conformations differ in their relative positions of the substrate-bound F-box protein FBXW7(ΔD) and ubiquitin transferase module, in both the ubiquitin-guided helix and Rcat domain of ARIH1 direct the C terminus of ubiquitin towards the F-box-protein-bound substrate. **e**, The catalytic architecture for TS2 is conserved irrespective of the TS2 ABP probe strategy. Cryo-EM map using TS2 ABP alternative (Extended Data Fig. 2e) is shown in grey, with docked model from corresponding SCF[FBXW7]–ARIH1(F430A/E431A/E503A)-ubiquitin-cyclin E complex captured with the TS2 cyclin E probe (**d** bottom left) (Fig.1d). The low-pass-filtered cryo-EM map obtained with the cyclin-E-based TS2 is shown in violet, superimposed on the right.

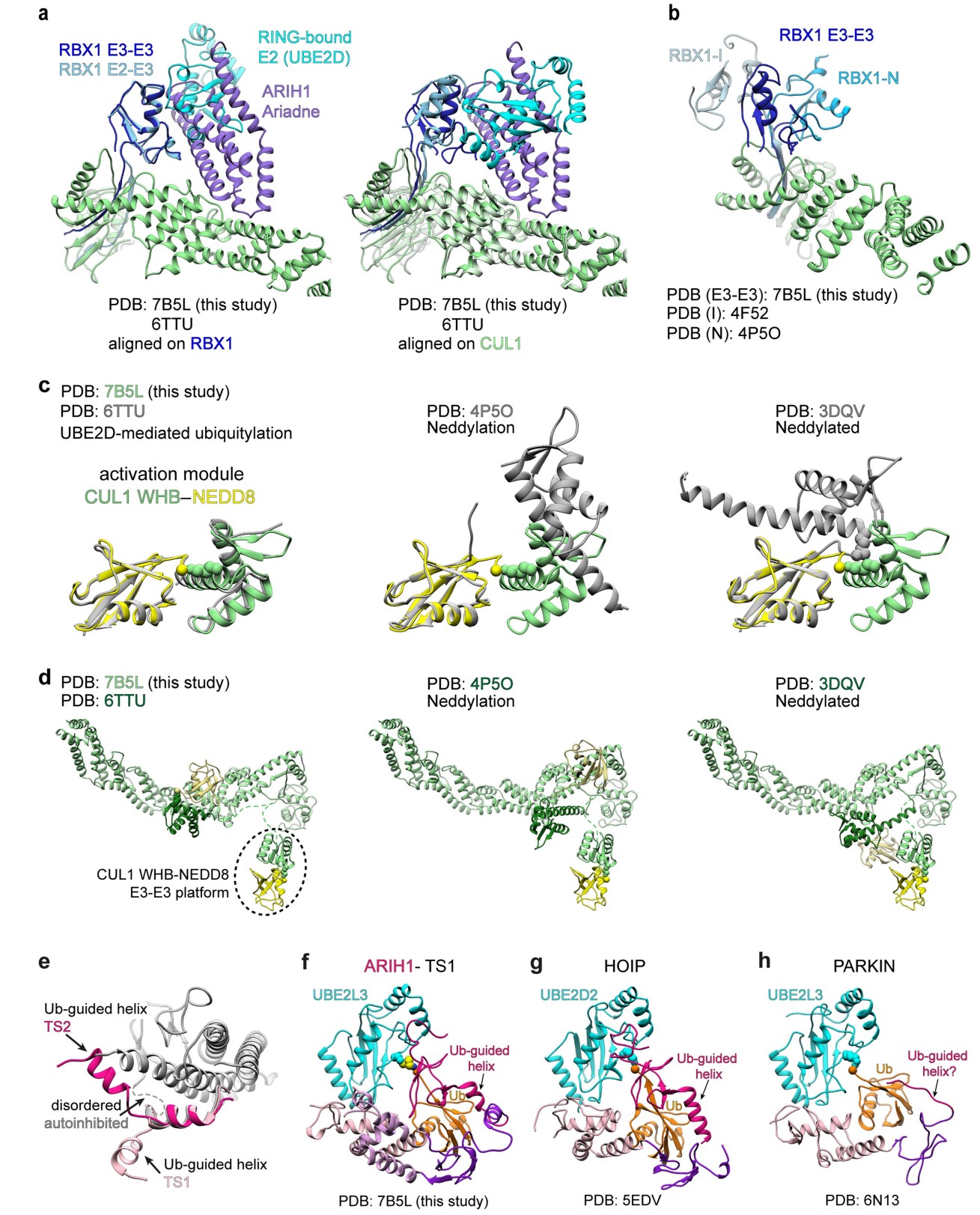

**a**
RBX1 E3-E3
RBX1 E2-E3
RING-bound E2 (UBE2D)
ARIH1 Ariadne

PDB: 7B5L (this study)
6TTU
aligned on RBX1

PDB: 7B5L (this study)
6TTU
aligned on CUL1

**b**
RBX1-I
RBX1 E3-E3
RBX1-N

PDB (E3-E3): 7B5L (this study)
PDB (I): 4F52
PDB (N): 4P5O

**c**
PDB: 7B5L (this study)
PDB: 6TTU
UBE2D-mediated ubiquitylation

activation module
CUL1 WHB–NEDD8

PDB: 4P5O
Neddylation

PDB: 3DQV
Neddylated

**d**
PDB: 7B5L (this study)
PDB: 6TTU

CUL1 WHB-NEDD8
E3-E3 platform

PDB: 4P5O
Neddylation

PDB: 3DQV
Neddylated

**e**
Ub-guided helix
TS2
disordered
autoinhibited
Ub-guided helix
TS1

**f**
ARIH1- TS1
UBE2L3
Ub-guided helix
Ub
PDB: 7B5L (this study)

**g**
HOIP
UBE2D2
Ub-guided helix
Ub
PDB: 5EDV

**h**
PARKIN
UBE2L3
Ub
Ub-guided helix?
PDB: 6N13

**Extended Data Fig. 5** | See next page for caption.

**Extended Data Fig. 5 | Structural rearrangements and implications of super-assembly for neddylated SCF and RBR E3 ligases. a**, Close-ups showing a common RBX1 RING surface binding to ARIH1 Ariadne domain in the E3–E3act super-domain and binding to the E2 UBE2D[17] (PDB 6TTU). Structures on left align CUL1–RBX1–ARIH1 and CUL1–RBX1–UBE2D over RBX1, and on right over CUL1. **b**, Unique RBX1 RING (blue) and CUL1 (green) arrangement in E3–E3act domain. For reference, the relative RBX1 RING orientations are shown for neddylation[51] in teal (RBX1 N, PDB 4P5O), and in an inhibited complex[69] in sky blue (RBX1I, PDB 4F52). **c**, Close-ups comparing relative orientations of NEDD8 and the isopeptide-linked WHB domain of CUL1 in TS1 (yellow–green) with the 'activation module' activating conventional UBE2D-dependent ubiquitylation of an SCF$^{\beta\text{-TRCP}}$ substrate[17] (PDB 6TTU) (grey) (left), or during neddylation[51] (PDB 4P5O) (grey) (middle), or captured by crystal packing in a structure of neddylated CUL5–RBX1 that revealed orientational flexibility of neddylated CUL WHB and RBX RING domains[16] (PDB 3DQV) (grey) (right). NEDD8 is superimposed across the different structures. **d**, Relative to CUL1 scaffold, NEDD8 (yellow)-linked CUL WHB domain (dark green) position in TS1 compared to positions of these domains from structures shown in **c**. After superimposing the CUL–RBX C/R domains, positions of NEDD8 are shown in light yellow and of its linked CUL1 WHB domain in dark green from structure representing UBE2D-dependent ubiquitylation of an SCF$^{\beta\text{-TRCP}}$ substrate (PDB 6TTU) (left), or during neddylation (PDB 4P5O) (middle) or captured by crystal packing in a structure of neddylated CUL5–RBX1 that revealed orientational flexibility of neddylated CUL WHB and RBX RING domains (PDB 3DQV) (right). Dotted lines show regions of structures connected but not modelled owing to lack of density. **e**, Comparison of structure and location of ARIH1 element corresponding to Ub-guided helix in TS1 (light pink) and TS2 (dark pink) relative to that linking the Ariadne and Rcat domains in the autoinhibited configuration[34] (grey). For perspective, the helix preceding the Ub-guided helix, which is similar in all of the structures, is also coloured. **f**, In the context of transient E3–E3 assembly, neddylated SCF-activated UBE2L3-ubiquitin-ARIH1 TS1 adopts the canonical activated RBR configuration. **g**, Corresponding region of structure with RBR E3 HOIP[41] (PDB 5EDV). **h**, Corresponding region of structure with PARKIN[45] (PDB 6N13). The structure of HOIP is represented as a monomer, although it is a domain-swapped dimer in the crystal. A HOIP ubiquitin-guided helix was previously noted to promote ubiquitin transfer from an E2 to this RBR E3 ligase[41]. This concept could possibly apply in PARKIN as well.

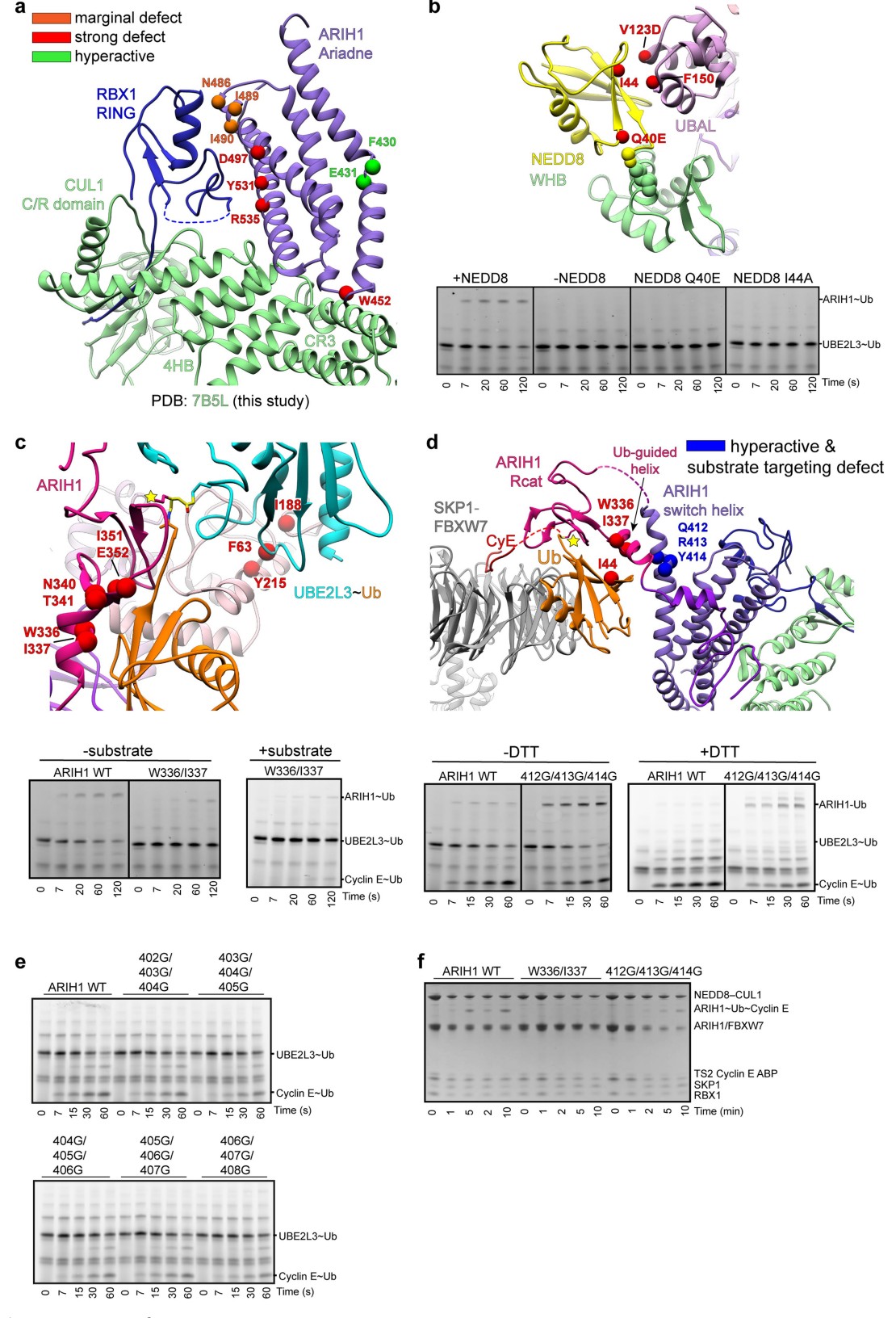

**Extended Data Fig. 6** | See next page for caption.

**Extended Data Fig. 6 | Mutational validation of structural mechanism of neddylated SCF–ARIH1-catalysed ubiquitylation. a**, Close-up of E3–E3act domain with spheres showing locations of strongly defective (red), marginally defective (orange) and hyperactive (green) mutations identified by previous ARIH1 Ala scanning mutagenesis[3,4]. Defective mutants map to key CUL1- and RBX1-binding residues, whereas hyperactive mutants map to the site of activating bend-to-kink conformation change within the switch helix. **b**, Close-up of CUL1-WHB-domain-linked NEDD8 interactions with ARIH1 UBAL domain in TS1, showing locations of strongly defective mutants as red spheres. Mutants in ARIH1 UBAL domain (V123D or F150A) at interface with NEDD8 were previously described[4]. A representative SDS–PAGE gel of experiments testing effects of mutating NEDD8 residues at interface with ARIH1 or with CUL1 in the context of neddylated CUL1–RBX1-activated ubiquitin transfer from UBE2L3 to ARIH1 is shown below. Gel image is representative of independent technical replicates ($n = 2$). **c**, Close-up of catalytic elements for TS1 (ubiquitin transfer from E2 UBE2L3 to ARIH1 catalytic cysteine (yellow star)). Red spheres show sites of previously identified strongly defective mutants[4], or tested on the basis of the structure representing TS1. SDS–PAGE gel (left) shows neddylated CUL1–RBX1-activated ubiquitin transfer from UBE2L3 to ARIH1 or (right) from UBE2L3 via ARIH1 to phosphopeptide substrate derived from cyclin E, testing

effects of ARIH1 mutants in the ubiquitin-guided helix preceding the Rcat domain. These residues are markedly remodelled for TS1 and were not tested in the previous ARIH1 Ala scanning mutagenesis study[4]. Gel images are representative of independent technical replicates ($n = 2$). **d**, Close-up of catalytic elements for TS2, ubiquitin transfer from ARIH1 to SCF-bound substrate. Spheres indicate sites of mutation defective in achieving substrate targeting configuration. Sites of blue mutations map to region in switch helix contributing to ARIH1 autoinhibition and substrate targeting, and accordingly lead to accumulation of ARIH1-ubiquitin in assays monitoring fluorescent ubiquitin transfer from UBE2L3 to ARIH1 to a neddylated SCF substrate (non-reducing conditions, −DTT (botttom left); reducing conditions, +DTT (bottom right)). Gel images are representative of independent technical replicates ($n = 2$). **e**, SDS–PAGE gels monitoring fluorescent ubiquitin transfer to cyclin E phosphopeptide, for ARIH1 triple-glycine mutants probing roles of the N-terminal end of the switch helix of the Ariadne domain. The mutations are upstream of the triple-glycine mutants shown as blue spheres in **d**. Gel images are representative of independent technical replicates ($n = 3$). **f**, SDS–PAGE gels of assays testing whether mutants adopt configuration for ubiquitin transfer from ARIH1 to substrate, probed by reaction with cyclin-E-based TS2 ABP. Gel image is representative of independent technical replicates ($n = 2$).

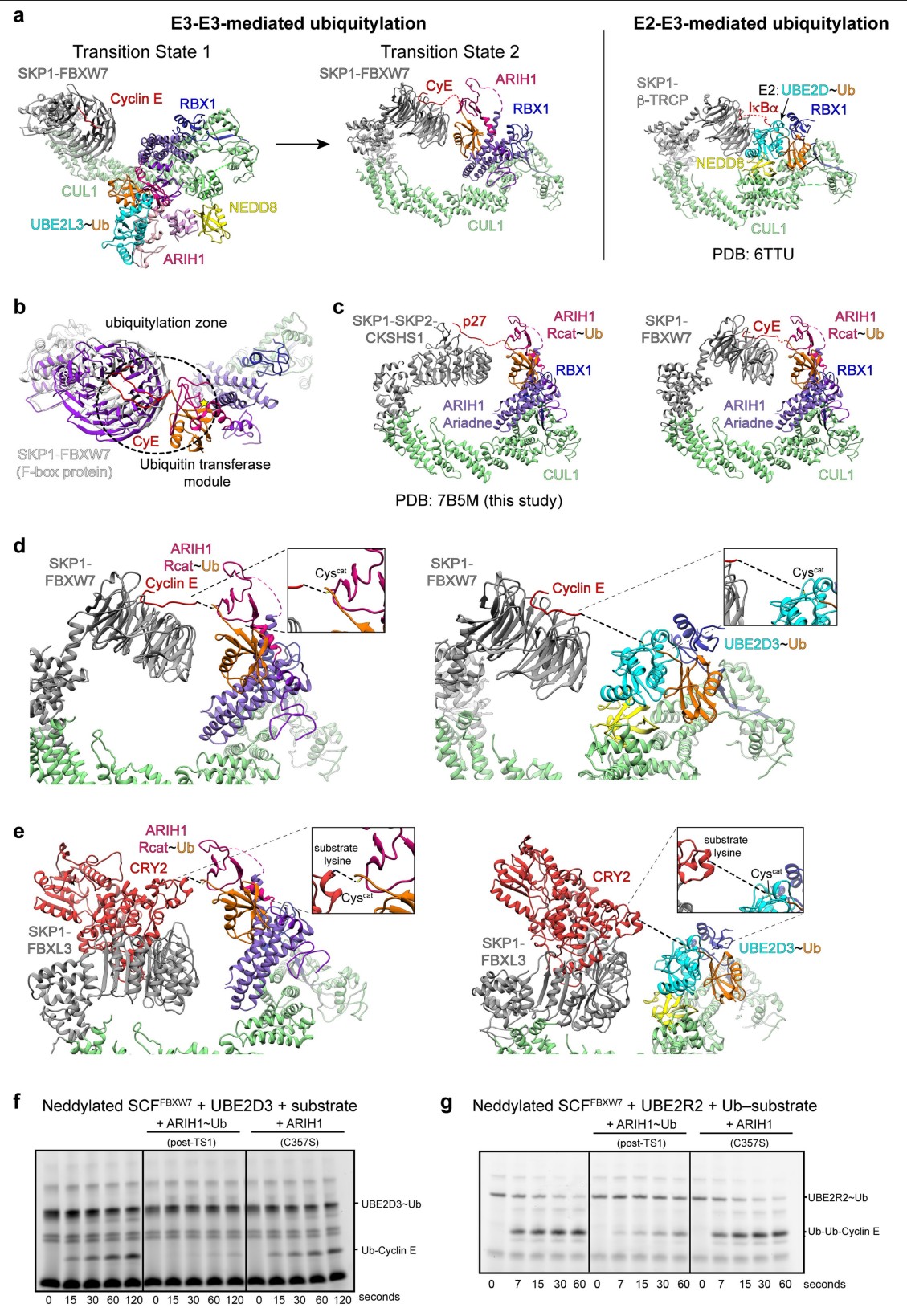

**Extended Data Fig. 7** | See next page for caption.

**Extended Data Fig. 7 | Comparison of structural mechanisms of neddylated SCF substrate ubiquitylation by ARIH1 (E3–E3) or conventional ubiquitylation with UBE2D (E2–E3). a**, Side-by-side comparison of E3–E3-catalysed substrate ubiquitylation, via the two transition states (left), versus conventional E2–E3 mechanism (right). Structures on left represent neddylated SCF$^{FBXW7}$-dependent ubiquitin transfer from UBE2L3 to ARIH1 (TS1) and from ARIH1 to cyclin E substrate (TS2) on the basis of electron microscopy data shown in Extended Data Fig. 4b, d. These structures are shown in two different relative orientations to highlight key features of TS1 or TS2 reactions. Structure on right shows neddylated SCF$^{β-TRCP}$-dependent ubiquitin transfer from the E2 UBE2D to IκBα[17] (PDB 6TTU). The structures are aligned over CUL1. In the E3–E3 mechanism, ubiquitin linked to the Rcat domain of ARIH1 is projected towards the substrate irrespective of F-box protein identity. However, in the E2–E3 mechanism, optimally positioned UBE2D specifically contacts β-TRCP, contributing to the notable catalytic efficiency of the conventional E2–E3 mechanism for neddylated SCF$^{β-TRCP}$ ubiquitylation of several of its substrates[17]. **b**, ARIH1-ubiquitin active site faces F-box-protein-bound substrate within a confined zone. Structures from two electron microscopy classes are shown (Extended Data Fig. 4d). After superimposing CUL1–RBX1–ARIH1-ubiquitin from both classes, the substrate (red)-bound F-box protein in one is shown in purple (conformation 1) and the other in grey (conformation 2). **c**, E3–E3 catalytic configuration is generalizable for substrates recruited to structurally diverse F-box proteins: p27 recruited to SKP1–SKP2–CKSHS1 (left) and cyclin E recruited to SKP1–FBXW7 (right). **d**, Structural modelling and comparison of E3–E3 versus E2–E3-mediated ubiquitylation with cyclin E as a substrate. The structure on the left corresponds to SKP1–FBXW7–cyclin E (PDB 2OVQ), fitted into map corresponding to conformation 1, with neddylated CUL1–RBX1-activated ARIH1-ubiquitin-substrate from the refined structure representing TS2 for SCF$^{SKP2}$. Proximity of the ubiquitin transferase domain to the substrate phospho-degron explains how ARIH1 efficiently ubiquitylates a 'short' cyclin E substrate, with only four residues between the phospho-degron and acceptor lysine (Fig. 3d, Extended Data Fig. 8). On the right is a model of E2–E3-mediated ubiquitylation of cyclin E by neddylated SCF$^{FBXW7}$. The model was generated by aligning the SKP1–F-box portion of SKP1–FBXW7–cyclin E (PDB 2OVQ) in place of the corresponding region showing UBE2D-mediated ubiquitylation of a substrate of neddylated SCF$^{β-TRCP}$ (PDB 6TTU), which shows the distance separating the catalytic cysteine of UBE2D and the cyclin E substrate acceptor. This rationalizes the inefficient ubiquitylation of the short cyclin E peptide substrate by the conventional E2–E3 mechanism (Fig. 3d, Extended Data Fig. 8). **e**, As in **d**, but modelled with SKP1–FBXL3–CRY2[70] (PDB 4I6J) based on the F-box proteins. **f**, Competition assay testing whether a neddylated SCF ligase can mediate conventional UBE2D-catalysed ubiquitylation if occupied by ARIH1. SDS–PAGE gels monitor neddylated SCF$^{FBXW7}$-dependent transfer of fluorescent ubiquitin from UBE2D3 to cyclin E phosphopeptide substrate. Ubiquitylation is severely hindered upon addition of the stable proxy for the E3–E3 post-TS1 intermediate (ARIH1-ubiquitin, generated by ARIH1 reaction with ubiquitin–VME) to the reaction, but not when ARIH1 is added on its own (right). The results are rationalized by the same portions of RBX1 and NEDD8 binding ARIH1 and UBE2D (Extended Data Fig. 5a, c). A catalytically inactive mutant version of ARIH1 in which the catalytic cysteine is substituted with Ser was used to prevent any potential spurious activity from ARIH1. Gel image is representative of independent technical replicates (*n* = 2). **g**, SDS–PAGE gels monitor neddylated SCF$^{FBXW7}$-dependent transfer of fluorescent ubiquitin from UBE2R2 to ubiquitin-linked cyclin E phosphopeptide substrate, testing competition upon adding the stable proxy for the E3–E3 post-TS1 intermediate (ARIH1-ubiquitin) to the reaction, or inactive ARIH1 on its own (right). Gel image is representative of independent technical replicates (*n* = 2).

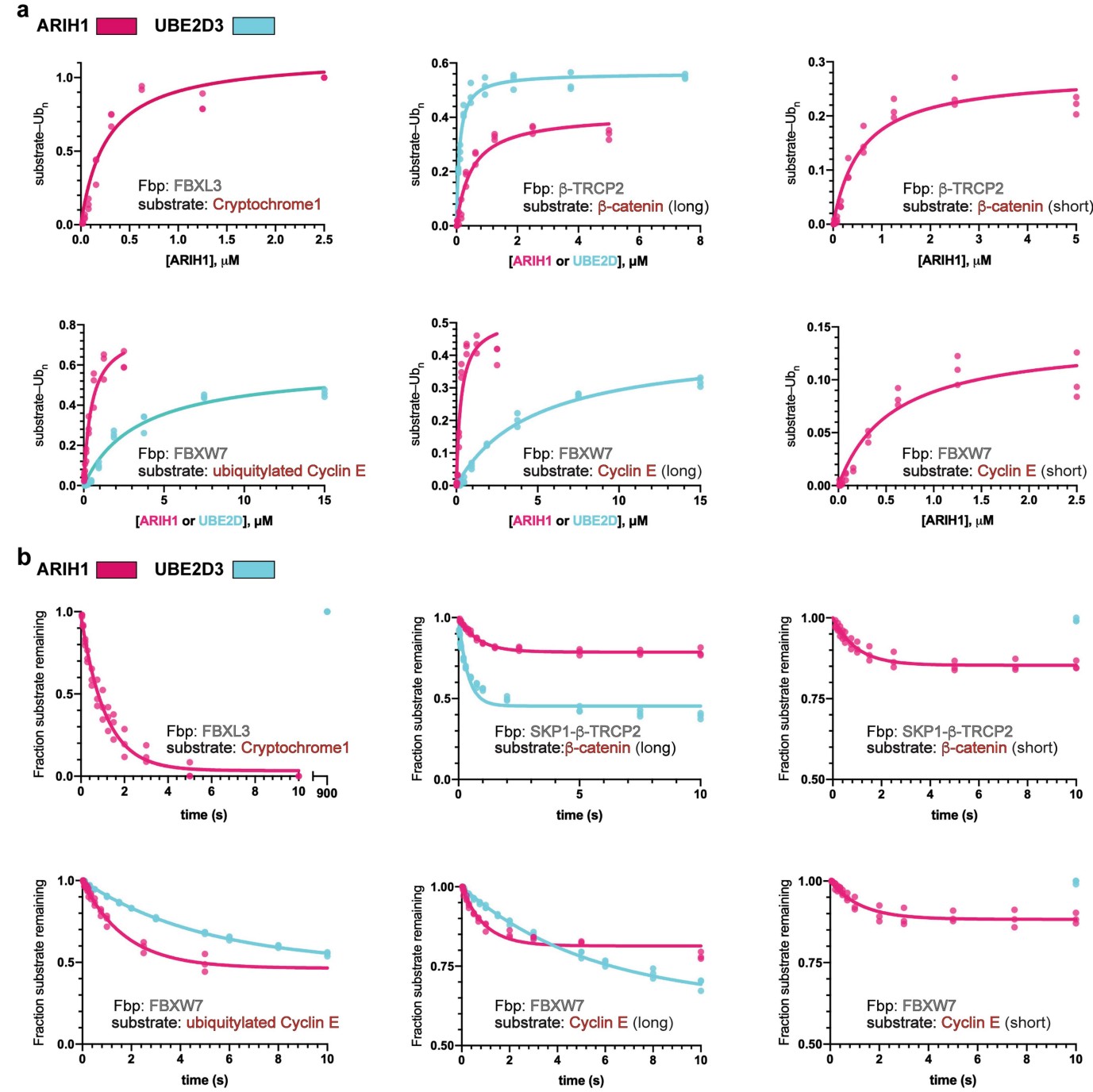

**Extended Data Fig. 8 | Quantitative kinetics show ARIH1 efficiently ubiquitylates diverse SCF-bound substrates, whereas UBE2D3 activity is relatively specialized. a**, Plots of the fraction of substrate that had been converted to ubiquitylated products versus concentration of ARIH1 (magenta) or UBE2D3 (cyan). Various neddylated SCF complexes were assayed that contained the F-box proteins FBXL3, β-TRCP2 (referred to throughout the Article as β-TRCP) or FBXW7 (ΔD version) and substrates cryptochrome 1 (CRY1) (relative position of substrate fixed by its folded structure), phosphopeptides derived from β-catenin (long (containing a lysine in a known substrate position[17]) or short (separated from the degron by only a four-residue linker)) or phosphopeptides derived from cyclin E (long (containing a lysine in a known substrate position[47]), short (separated from the degron by only a four-residue linker) or ubiquitylated (contains a single ubiquitin fused by sortase-mediated transpeptidation)). Triplicate data points from independent experiments are shown and were fit to the Michaelis–Menten model to estimate the $K_m$ of ARIH1 or UBE2D3 for the respective neddylated SCF–substrate complexes using nonlinear curve fitting (GraphPad Prism). **b**, Plots comparing various neddylated SCF ligases and substrates described in **a**, showing appearance of ubiquitylated substrate (CRY1) or disappearance of unmodified substrate in rapid-quench flow reactions all performed under single-encounter conditions. Triplicate data points from independent experiments are shown. The data were fit to closed form equations (Mathematica) as previously described[58] to obtain rates of ubiquitylation ($k_{obs}$) as well as their associated standard error (Extended Data Table 2).

**Extended Data Table 1 | Cryo-EM data collection, refinement and validation statistics**

| | Pre-TS1 | TS1 | TS1 | post-TS1 | TS2 | TS2 | TS2 |
|---|---|---|---|---|---|---|---|
| Intermediate | Pre-TS1 | TS1 | TS1 | post-TS1 | TS2 | TS2 | TS2 |
| ABP | none | TS1 ABP | TS1 ABP | Ub-VME | TS2 CyE ABP | TS2 CyE ABP | TS2 p27 ABP |
| Substrate Receptor | FBXW7$^{263\text{-}C}$ | SKP2 | FBXW7$^{263\text{-}C}$ | FBXW7$^{263\text{-}C}$ | FBXW7$^{263\text{-}C}$ | FBXW7$^{263\text{-}C}$ | SKP2 |
| ARIH1 | WT | WT | WT | WT | 90-C WT | 90-C* | 90-C* |
| NEDD8 | yes | yes | yes | yes | no | yes | no |
| additional | isopeptide-linked UBE2L3~Ub Cyclin E peptide | Cyclin A$^{170\text{-}C}$, CDK2 CKSHS1$^{5\text{-}73}$, p27$^{22\text{-}106}$ p27 phosphopeptide | Cyclin E phosphopep. | Cyclin E phosphopep. | | | CKSHS1$^{5\text{-}73}$ |
| | | EMD-12037 (composite) EMD-12038 (consensus) EMD-12048 (focused1 CyA/CDK) EMD-12041 (focused2 catalytic) EMD-12050 (focused3 Cullin) | | | | | |
| | EMD-12004 | | EMD-12036 | EMD-12005 | EMD-12006 | EMD-12039 | EMD-12040 |
| **Data collection and processing** | | | | | | | |
| Microscope | Krios | Krios | Arctica | Glacios | Krios | Krios | Krios |
| Magnification | 105,000 | 105,000 | 73,000 | 22,000 | 130,000 | 130,000 | 105,000 |
| Voltage (kV) | 300 | 300 | 200 | 200 | 300 | 300 | 200 |
| Electron exposure(e–/Å$^2$) | 70 | 70 | 60 | 70 | 69 | 69 | 79 |
| Defocus range (μm) | -0.8 ~ -3.3 | -0.8 ~ -3.3 | -1.5 ~ -3.5 | -0.8 ~ -3.3 | -1.2 ~ -3.3 | -1.2 ~ -3.3 | -1.2 ~ -3.3 |
| Pixel size (Å) | 0.851 | 1.09 | 1.997 | 1.181 | 0.851 | 0.851 | 1.09 |
| Symmetry | C1 | C1 | C1 | C1 | C1 | C1 | C1 |
| Initial particle images (no.) | 2,651,495 | 5,467,024 | 651,656 | 723,348 | 2,525,317 | 2,568,237 | 2,709,623 |
| Final particle images (no.) | 122,649 | 623,409 | 130,116 | | 462,256 | 151,246 | 759,489 |
| Map resolution (Å) | 4.5 | 3.8 | 9.6 | | 3.6 | 4.4 | 3.9 |
| FSC threshold | (0.143) | (0.143) | (0.143) | | (0.143) | (0.143) | (0.143) |
| **Refinement** | | composite | focused1 | focused2 | focused3 | | |
| PDB code: | | 7B5L | 7B5R | 7B5N | 7B5S | | 7B5M |
| Initial model used (PDB code) | | 1LDJ 5UDH 2AST 6TTU 1H27 | 1H27 1LDJ 2AST | 5UDH 1LDJ 6TTU | 1LDJ5 UDH | | 1LDJ 5UDH 2AST |
| Model resolution (Å) | | 3.6 | 3.6 | 3.8 | 3.6 | | 4.0 |
| FSC threshold | | (0.143) | (0.143) | (0.143) | (0.143) | | (0.143) |
| Model composition | | | | | | | |
| Non-hydr. Atoms | | 21529 | 11606 | 9965 | 7671 | | 12526 |
| Protein residues | | 2654 | 1430 | 1229 | 931 | | 1561 |
| Ligands | | 9 ZN | | 9 ZN | 5 ZN | | 5 ZN |
| $B$ factors (Å$^2$) | | | | | | | |
| Protein | | 134.15 | 136.68 | 63.66 | 68.70 | | 93.74 |
| Ligand | | 158.66 | | 99.15 | 122.41 | | 184.43 |
| R.m.s. deviations | | | | | | | |
| Bond lengths (Å) | | 0.006 | 0.010 | 0.007 | 0.006 | | 0.009 |
| Bond angles (°) | | 0.752 | 1.002 | 0.794 | 0.772 | | 0.966 |
| Validation | | | | | | | |
| MolProbity score | | 2.23 | 2.50 | 2.16 | 2.01 | | 2.55 |
| Clashscore | | 15 | 16 | 13 | 10 | | 14 |
| Poor rotamers (%) | | | | | | | |
| Ramachandran plot | | | | | | | |
| Favored (%) | | 90 | 90 | 91 | 91 | | 90 |
| Allowed (%) | | 10 | 10 | 9 | 9 | | 10 |
| Disallowed (%) | | 0.00 | 0 | 0 | 0 | | 0 |

**Extended Data Table 2 | Estimates of $K_m$ and $k_{obs}$ for substrate ubiquitylation**

| Substrate | E3 or E2 | SCF | $K_m$ ($10^{-9}$ M) | $k_{obs}$ (sec$^{-1}$) | $k_{obs}/K_m$ (M$^{-1}$sec$^{-1}$) |
|---|---|---|---|---|---|
| Cyclin E | ARIH1 | FBXW7 | 288 ± 68 | 0.25 ± 0.026 | 8.7x10$^5$ |
| Cyclin E | UBE2D3 | FBXW7 | 4789 ± 529 | 0.08 ± 0.002 | 1.7x10$^4$ |
| ubiquitylated Cyclin E | ARIH1 | FBXW7 | 489 ± 82 | 0.39 ± 0.008 | 8.0x10$^5$ |
| ubiquitylated Cyclin E | UBE2D3 | FBXW7 | 3416 ± 545 | 0.13 ± 0.002 | 3.8x10$^4$ |
| Cyclin E short | ARIH1 | FBXW7 | 568 ± 137 | 0.12 ± 0.009 | 2.1x10$^5$ |
| Cyclin E short | UBE2D3 | FBXW7 | NA | NA | NA |
| β-catenin | ARIH1 | β–TRCP2 | 499 ± 88 | 0.41 ± 0.012 | 8.2x10$^5$ |
| β-catenin | UBE2D3 | β–TRCP2 | 100 ± 9 | 2.2 ± 0.29 | 2.2x10$^7$ |
| β-catenin short | ARIH1 | β–TRCP2 | 590 ± 103 | 0.24 ± 0.007 | 4.1x10$^5$ |
| β-catenin short | UBE2D3 | β–TRCP2 | NA | NA | NA |
| CRY1 | ARIH1 | FBXL3 | 265 ± 60 | 0.90 ± 0.069 | 3.4x10$^6$ |
| CRY1 | UBE2D3 | FBXL3 | NA | NA | NA |

# nature research

# Reporting Summary

Nature Research wishes to improve the reproducibility of the work that we publish. This form provides structure for consistency and transparency in reporting. For further information on Nature Research policies, see Authors & Referees and the Editorial Policy Checklist.

## Statistics

For all statistical analyses, confirm that the following items are present in the figure legend, table legend, main text, or Methods section.

| n/a | Confirmed | |
|---|---|---|
| ☐ | ☒ | The exact sample size (*n*) for each experimental group/condition, given as a discrete number and unit of measurement |
| ☐ | ☒ | A statement on whether measurements were taken from distinct samples or whether the same sample was measured repeatedly |
| ☒ | ☐ | The statistical test(s) used AND whether they are one- or two-sided *Only common tests should be described solely by name; describe more complex techniques in the Methods section.* |
| ☒ | ☐ | A description of all covariates tested |
| ☒ | ☐ | A description of any assumptions or corrections, such as tests of normality and adjustment for multiple comparisons |
| ☒ | ☐ | A full description of the statistical parameters including central tendency (e.g. means) or other basic estimates (e.g. regression coefficient) AND variation (e.g. standard deviation) or associated estimates of uncertainty (e.g. confidence intervals) |
| ☒ | ☐ | For null hypothesis testing, the test statistic (e.g. *F*, *t*, *r*) with confidence intervals, effect sizes, degrees of freedom and *P* value noted *Give P values as exact values whenever suitable.* |
| ☒ | ☐ | For Bayesian analysis, information on the choice of priors and Markov chain Monte Carlo settings |
| ☒ | ☐ | For hierarchical and complex designs, identification of the appropriate level for tests and full reporting of outcomes |
| ☒ | ☐ | Estimates of effect sizes (e.g. Cohen's *d*, Pearson's *r*), indicating how they were calculated |

*Our web collection on statistics for biologists contains articles on many of the points above.*

## Software and code

Policy information about availability of computer code

Data collection | Gel imaging: Amersham Imager 600, Amersham Typhoon;  Cryo-EM: SerialEM, FEI EPU

Data analysis | Kinetics: ImageQuant v5.2, Mathematica v10.0.0, GraphPad Prism v8.2.1 Cryo-EM: RELION 3.1, Gautomatch v.056, Gctf v1.06;  Structure Visualization: Chimera 1.14, ChimeraX 1.0, PyMol 1.5.0.4;  Model Building: COOT 0.8.9.2, Phenix.refine 1.18.2

For manuscripts utilizing custom algorithms or software that are central to the research but not yet described in published literature, software must be made available to editors/reviewers. We strongly encourage code deposition in a community repository (e.g. GitHub). See the Nature Research guidelines for submitting code & software for further information.

## Data

Policy information about availability of data

All manuscripts must include a data availability statement. This statement should provide the following information, where applicable:
- Accession codes, unique identifiers, or web links for publicly available datasets
- A list of figures that have associated raw data
- A description of any restrictions on data availability

EMDB-12004,12005, 12006, 12007, 12036, 12037, 12038, 12040, 12041, 12048, 12050. PDB-7B5L, 7B5M, 7B5N, 7B5R, 7B5S

# Field-specific reporting

Please select the one below that is the best fit for your research. If you are not sure, read the appropriate sections before making your selection.

☒ Life sciences  ☐ Behavioural & social sciences  ☐ Ecological, evolutionary & environmental sciences

# Life sciences study design

All studies must disclose on these points even when the disclosure is negative.

| | |
|---|---|
| Sample size | All experiments we performed at least twice. |
| Data exclusions | No data were excluded. |
| Replication | All experiments were performed at least twice, with numerous controls. All attempts at replication were successful. |
| Randomization | No grouped samples. |
| Blinding | No grouped samples. |

# Reporting for specific materials, systems and methods

We require information from authors about some types of materials, experimental systems and methods used in many studies. Here, indicate whether each material, system or method listed is relevant to your study. If you are not sure if a list item applies to your research, read the appropriate section before selecting a response.

## Materials & experimental systems

| n/a | Involved in the study |
|---|---|
| ☒ | ☐ Antibodies |
| ☒ | ☐ Eukaryotic cell lines |
| ☒ | ☐ Palaeontology |
| ☒ | ☐ Animals and other organisms |
| ☒ | ☐ Human research participants |
| ☒ | ☐ Clinical data |

## Methods

| n/a | Involved in the study |
|---|---|
| ☒ | ☐ ChIP-seq |
| ☒ | ☐ Flow cytometry |
| ☒ | ☐ MRI-based neuroimaging |

