## [Peer Review File · Nature]

Manuscript Title: Ubiquitin ligation to F-box protein targets by SCF-RBR E3-E3 super-assembly

Reviewer Comments & Author Rebuttals

Reviewer Reports on the Initial Version:

Referee expertise:

Referee #1: Cryo-EM, ubiquitin biology

Referee #2: Structural and functional aspects of ubiquitin signalling

Referee #3: Structural aspects of ubiquitin signalling

Referees' comments:

Referee #1 (Remarks to the Author):

SCF E3 ligases regulate multiple diverse cellular processes. An interesting variant is the combination of SCF with the RBR E3 ligase ARIH1 to mediate SCF-bound substrates by means of an E3-E3 ligase assembly. The rate of ubiquitination of SCF-bound substrates by the E3-E3 ligase assembly is apparently more efficient than SCF-mediated ubiquitination. Thus far, the structural basis of this catalytic reaction is unknown.

The manuscript by Schuman and colleagues combines chemical biology, biochemistry and cryo-EM to characterise two transition state reaction intermediates involved in the process by which the SCF E3 ligase controls the ubiquitination of SCF-bound substrates by the E2-RBR complex Ube2L3-ARIH1. Using transition state mimics of the two reactions (TS1: ARIH1 ubiquitination by Ube2L3, and TS2: ubiquitination of F-box substrates of SCF by ARIH1~Ub), the cryo-EM structures of a total of seven complexes were determined, allowing analysis of different F-box substrates. The results collectively explain how SCF promotes ubiquitination of ARIH1 by relieving an auto-inhibitory state of ARIH1, and then how SCF promotes the ubiquitination of its target substrates. Mechanistically this is mediated by large conformational changes of the assembly.

Overall this is an important study that provides significant insights into the mechanism of the E3-E3-assembly catalytic mechanism, explaining how these multi-domain complex enzymes interact.

There are a few points for the authors to consider listed below. As a general point the manuscript is in places difficult to follow. This is due to (i) incomplete figure labelling, (ii) the inconsistent naming of complexes in text, figures and Extended Table 1, (iii) reference to non-existent figures. The authors should carefully review all figures to ensure they are comprehensible and that complexes are named in a consistent manner throughout the manuscript.

1. The authors claim that their results explain why the SCF is efficient at ubiquitinating b-TRCP-bound substrates, but that ARIH1 enhances the SCF-mediated ubiquitination of other F-box-dependent proteins (mix-and-match) (pages, 4 and 11, Fig. 4d). I wasn't clear what the authors' rationale for this is and how this was tested.

2. The authors use a clever chemical approach to trap transition state intermediates to visualise the reaction process of RBR ubiquitination by the E2 (TS1) and substrate ubiquitination by the E3-E3 assembly. However, these chemical mimics do not exactly match the chemical configuration of the transition state, particularly for the TS1 mimic where the three-way junction between E2, Ub and Arh1 is such that the E2 Cys S-atom is displaced by 3 atoms in the TS intermediate relative to the actual transition state. The authors show that the activity based probe (ABP) reacts as expected, but it would be of interest to understand how the kinetics of formation of the TS1 state differs relative to ARIH1 ubiquitination by Ube2L3~Ub (Fig. 1b-c). Similarly, for the rate transfer of Ub from the super E3-E3

enzyme to substrate compared to the rate of the TS2 formation (Fig. 1e-g).

3. The authors have determined the structures of 7 complexes (Extended Data Table 1), but the naming of these complexes does not seem consistent throughout the text. For example, is the complex in Extended Data Fig. 2 the same as in Extended Data Fig. 3a-d? and then in Extended Data 4a.

Understandably the full name of the complex with all subunits is long, but the abbreviations used by the authors are ambiguous. The authors should consider giving each complex a unique abbreviation used when in the text, figures and tables.

4. The authors use the symbol '>' repeatedly in different contexts but never for the correct meaning of 'greater than'. This should be corrected because it is confusing.

5. Figures 1d, 1h are very difficult to understand. There is a mismatch between the colours used in the cryo-EM reconstruction (a lot of grey) and the schematic above of ARIH1. The implication is the grey domains in the ARIH1 schematic are grey in the reconstruction, but this presumably is not the case because other parts of the EM reconstruction correspond to SCF. The EM reconstruction figure 1d, 1h (and others) need much more careful labelling.

6. Fig. 3d: labelling unclear. RTI helix should be labelled.

7. Fig. 4b. All subunits should be labelled.

8. Where is the ubiquitinase module defined?

9. Page 4. '.. penultimate Gly75..' doesn't make sense. It is either '.. penultimate Gly..' or '.. Gly75...'.

10. Page 5. ARIH1* isn't defined.

11. Page 5. Figure 1i (twice) doesn't exist, presumably Figure 1h.

12. Page 6. What does 'evolving catalytic configurations' mean? Not in the genetic sense presumably or is it?. The use is ambiguous.

13. Page 6. Reference needed for the statement in the first line.

14. Page 8. I can't see p27 in Figure 2b.

Referee #2 (Remarks to the Author):

In the manuscript, "Ubiquitin ligation to F-box protein-bound substrates by transient RING-RBR E3-E3 ligase superassembly", Schulman and colleagues provide structural details of how CRL-ARIH E3-E3 ligases mediate ubiquitin transfer to targets. CRLs represent the largest fraction of known E3 ligases and play essential roles in many areas of biology. In 2016, this group (Scott et al.) identified ARIH RBR E3 ligases as important components of some CRL ubiquitination reactions through forming CRL-ARIH E3-E3 ligase superassemblies. However, many questions remained from this work, including how these large complexes with many parts (some seemingly redundant) coordinate ubiquitin transfer from E2 to target. Notably, this work also follows up on a recent publication from the Schulman group this year (Baek et al.) that again defined structural details of how a "specialist" CRL-E2 assembly mediates ubiquitin transfer to substrates. This current study essentially marries these two previous studies through a series of impressive cryo-EM structures that track ubiquitin transfer through the enzymatic reaction from E2 -> RBR -> target. Two transition state mimics are captured that allow understanding of this process and how large structural rearrangements are induced.

Although this work is a masterpiece in structural biology, it is not clear to this reviewer how much closer

it brings us to understanding the central questions that have been around ever since the '2 E2' model was first suggested and were accentuated by Scott et al. Below are three key questions related to this.

Major points to be addressed by the authors:

1) What, if anything, determines whether an SCF attempts to go down the trajectory of direct transfer from E2 to substrate or via ARIH1? Is it completely stochastic? For example, does an SCFb-TrCP sometimes engage ARIH1 for priming? Does SCFSkp2 bound to p27 sometimes first recruit an E2~Ub but then the E2~Ub dissociates before it can manage to discharge its ubiquitin and priming does not occur until ARIH1 is recruited? The authors hint at potential "specialist" CRLs that may not rely on ARIH based on interaction between E2 and substrate adaptor. This includes their recent study (Baek et al.) on SCFb-TrCP-UBE2D. However, providing more generalizable insights into these questions would resoundingly improve the impact of the study.

2) How do the structural data known to date explain what appears to be a very strong tendency of ARIH1 to drive monoubiquitination? It would appear that once one ubiquitin is transferred, there is significant preference for a chain-extending E2 to take over (like Ube2R1/2 or Ube2g1/g2). Again, is there anything in the structural biology that yields clues as to how this might work, or is it completely stochastic and there is no mechanism to favor whether ARIH1 or E2 is recruited for either a naked or ubiquitylated substrate, and the system relies entirely on fast dynamics such that the residence time of ARIH1+Ube2D3~Ub or E2~Ub on Cul1-RING is short enough that the negative effects of competitive binding are negligible?

3) Whereas the work shown provides detailed insight into the structures of the sequential transition states, it has much less to say about how the system traverses from the first transition state to the second. One simple explanation is that the first transition state has pent-up potential energy, analogous to a compressed spring, and the instant that the bond linking Ub to Ube2D3 is severed, the system relaxes towards the second transition state. With this in mind, there is one analysis that I would have liked to see, which is to examine the structures using simpler traps that link the ubiquitin to either Ube2D3 or ARIH1. The latter would be particularly interesting with respect to the hypothesis above. The expectation is that the SCF+ARIH1+Ube2D3~Ub structure would look like TS1 whereas the SCF+ARIH1~Ub+Ube2D3 structure might look like TS2 (presumably the Ube2D3 would be absent).

Minor points:

- 1) Some of the figures may be difficult to interpret by a naïve audience. Addition of some models at key points may be helpful.
- 2) Figure 1h is referred to as Figure 1i in the text.

Referee #3 (Remarks to the Author):

The manuscript entitled "Ubiquitin ligation to F-box protein-bound substrates by transient RING-RBR E3-E3 ligase superassembly" by Horn-Ghetko et al., presents structures for several reconstituted complexes representing key intermediates in the transfer of ubiquitin from an E2 to an RBR E3 and from an RBR E3 to substrates bound to various flavors of reconstitute Cullin ring ligase complexes. The topic is timely, and it validates several hypotheses that were generated several years ago when it became clear that the ARIH RBRs cooperated with CRLs and NEDD8 to enhance substrate ubiquitination. The structures are all at modest resolution, but as many of the components have structures available through other means, the authors assemble a convincing array of models depicting nearly every step in this process. These structures show an array of interactions involving mutual exclusivity as well as composite surfaces that nicely illustrate mechanisms that alleviate inhibition or enforce directionality and specificity. And models explaining the dependence on Nedd8 (or lack thereof) are striking and also of significant note. This could not have been achieved without the clever use of several chemical strategies to bring normally reactive components together into stable complexes. While several of the strategies to link components together have some limitations with respect to spacing/chirality, it is the opinion of this reviewer that they are sufficient to trap relevant conformational intermediates in this process. The work is buttressed by

interpretation of prior or new mutational and biochemical analyses, and thus it represents a more complete body of work in comparison to many recently published cryoEM studies. There are only a few minor concerns that should be easily addressed by the authors.

If the authors have collected any more data in the interim, they are encouraged to improve the overall and local resolutions for the more important and relevant complexes. If this data does not exist, then the story should be published as is.

One major strength of this study is the expert use of chemical strategies to bring the various reactants together, yet electron densities and corresponding atomic models for the respective linkages are either not presented or hidden behind less interesting parts of the structure (Ext Fig 3D, 3H). Close-up views of the linkages and active sites must be shown. In addition, it is difficult to ascertain the respective limitations of the chemical crosslinking strategies with respect to chirality or the relative distances between active sites or Ub tail in linked or theoretical transition states. As stated earlier, it is clear that these strategies worked to bring the reactants together, but they may not capture all of the requisite elements of the reactions if the length or topologies differ from the true transition states. For instance, Fig 1b suggests that the distance between the E2 and E3 is greater in the mimic, but by how much? Would the interface between the E2 and E3 differ if it were shorter? The reader can do these calculations, but perhaps a S to S distance could be included? Similar point in Fig 1e, is the NH of the substrate indicating the primary amine of the N-terminus or a lysine side chain? If it's the latter than the distance between the substrate, Ub and E3 is quite a bit longer in the native reaction.

Fig 2 d,e could be improved, the overlay makes it difficult to see what is going on. Side by side might be clearer.

Terms such as 'immense' and 'massive' are subjective and should perhaps be removed from the abstract.

Author Rebuttals to Initial Comments:

General response to reviewers:

We thank the Reviewers for their encouraging comments and for many suggestions for improving our manuscript. To address your questions, we performed many new experiments and revised the text and figures. Responses are indicated in blue.

Referee #1 (Remarks to the Author):

SCF E3 ligases regulate multiple diverse cellular processes. An interesting variant is the combination of SCF with the RBR E3 ligase ARIH1 to mediate SCF-bound substrates by means of an E3-E3 ligase assembly. The rate of ubiquitination of SCF-bound substrates by the E3-E3 ligase assembly is apparently more efficient than SCF-mediated ubiquitination. Thus far, the structural basis of this catalytic reaction is unknown.

The manuscript by Schuman and colleagues combines chemical biology, biochemistry and cryo-EM to characterise two transition state reaction intermediates involved in the process by which the SCF E3 ligase controls the ubiquitination of SCF-bound substrates by the E2-RBR complex Ube2L3-ARIH1. Using transition state mimics of the two reactions (TS1: ARIH1 ubiquitination by Ube2L3, and TS2: ubiquitination of F-box substrates of SCF by ARIH1~Ub), the cryo-EM structures of a total of seven complexes were determined, allowing analysis of different F-box substrates. The results collectively explain how SCF promotes ubiquitination of ARIH1 by relieving an auto-inhibitory state of ARIH1, and then how SCF promotes the ubiquitination of its target substrates. Mechanistically this is mediated by large conformational changes of the assembly.

Overall this is an important study that provides significant insights into the mechanism of the E3-E3-assembly catalytic mechanism, explaining how these multi-domain complex enzymes interact.

We are very pleased by the Reviewer's enthusiasm for our work!

There are a few points for the authors to consider listed below. As a general point the manuscript is in places difficult to follow. This is due to (i) incomplete figure labelling, (ii) the inconsistent naming of complexes in text, figures and Extended Table 1, (iii) reference to non-existent figures. The authors should carefully review all figures to ensure they are comprehensible and that complexes are named in a consistent manner throughout the manuscript.

We thank the reviewer for these helpful comments. In the context of the revision, they are even more relevant as many new experiments were added. We have streamlined the names for the intermediates across the text and figures, and have taken care to reference correct figure panels.

1. The authors claim that their results explain why the SCF is efficient at ubiquitinating b-TRCP-bound substrates, but that ARIH1 enhances the SCF-mediated ubiquitination of other F-box-dependent proteins (mix-and-match) (pages, 4 and 11, Fig. 4d). I wasn't clear what the authors' rationale for this is and how this was tested.

This comment (and Major point #1 from Reviewer 2) made us realize that this concept has not actually been quantitatively tested. Thus, we addressed this with new analyses and experiments. First, we compared the structures representing the two different mechanisms for neddylated SCF substrate ubiquitylation, which revealed striking differences (Figure 3,

Extended Data Figure 7). The E3-E3 structures show the ARIH1~ubiquitin active site localized near substrates of diverse F-box proteins (SKP2 and FBXW7). Moreover, modeling another F-box protein (FBXL3) based on our cryo EM data showed its substrate, CRY1, could likewise approach the ARIH1~ubiquitin active site. The structural modeling suggested the E3-E3 mechanism is compatible with a range of geometries, including substrates proximal to an F-box protein, and also a ubiquitylated substrate. In contrast, models based on the prior structure representing ubiquitin transfer from the E2 UBE2D to a neddylation SCF $^{\beta}$ -TRCP-bound peptide substrate showed that the RBX1 RING activated UBE2D~ubiquitin active site is optimally placed for substrates of β -TRCP, in part due to UBE2D directly contacting β -TRCP. Moreover, the location of the UBE2D~ubiquitin active site appeared compatible with some substrates but not others.

Second, to quantitatively compare the E3-E3 and E3-E2 mechanisms, we determined kinetic parameters by rapid quench-flow methods (Figure 3, Extended Data Table 2). This confirmed broad utility of the E3-E3 mechanism, with relatively similar catalytic efficiencies for all tested neddylation SCFs and substrates, including chain elongation of a ubiquitin-modified substrate. The conventional E3-E2 mechanism with UBE2D was optimal for neddylation SCF $^{\beta}$ -TRCP and could ubiquitylate disordered peptide substrates of sufficient length to simultaneously engage an Fbp and the UBE2D~ubiquitin active site. However, ubiquitylation of substrates that cannot accommodate this E3-E2 catalytic geometry – either due to the substrate's folded structure, or to limited distance between an F-box protein binding phosphodegron and acceptor lysine was only quantifiable for ARIH1.

2. The authors use a clever chemical approach to trap transition state intermediates to visualise the reaction process of RBR ubiquitination by the E2 (TS1) and substrate ubiquitination by the E3-E3 assembly. However, these chemical mimics do not exactly match the chemical configuration of the transition state, particularly for the TS1 mimic where the three-way junction between E2, Ub and Arh1 is such that the E2 Cys S-atom is displaced by 3 atoms in the TS intermediate relative to the actual transition state. The authors show that the activity based probe (ABP) reacts as expected, but it would be of interest to understand how the kinetics of formation of the TS1 state differs relative to ARIH1 ubiquitination by Ube2L3~Ub (Fig. 1b-c). Similarly, for the rate transfer of Ub from the super E3-E3 enzyme to substrate compared to the rate of the TS2 formation (Fig. 1e-g).

We performed several experiments to address these questions. Because an entire neddylation SCF-ARIH1 reaction occurs in less than one second, it was necessary to use a rapid quench-flow device to resolve the intermediates. This is now shown in Extended Data Figure 1b and c. We also monitored time-courses of TS1 and TS2 probe reactions under similar conditions in Extended Data Figure 1e and f. While the probes are slower, in both cases some product was observed in minutes and product increased over time.

We designed our TS1 and TS2 probes to match the atomic distance between ubiquitin and the RBR active site during the transition states, rationalizing that the distance between ubiquitin and E2 in the TS1 probe is in the same range as that in many structures employing isopeptide-bonded mimics of E2~ubiquitin intermediates, and that the flexibility of peptide substrates should enable achieving an overall configuration representing the TS2 intermediate. In general, reactions with E3 ligase probes reported to date are far slower than native reactions, often taking hours or overnight (Pao et al., Nature Chem Bio 2016 and Nature 2018). Thus, although the reactions with our ABPs proceed more slowly than the native ubiquitylation reactions, the rates seem at least on par with other E3 ligase probes.

Due to the Reviewer's questions about the TS1 reaction, we also performed several additional controls. We now show that reactivity of the TS1 probe requires CUL1-RBX1 neddylation, and is thwarted by ARIH1 mutations that would impair binding to NEDD8, CUL1, and RBX1. These new data are now included in Extended Data Figure 2d.

3. The authors have determined the structures of 7 complexes (Extended Data Table 1), but the naming of these complexes does not seem consistent throughout the text. For example, is the complex in Extended Data Fig. 2 the same as in Extended Data Fig. 3a-d? and then in Extended Data 4a. Understandably the full name of the complex with all subunits is long, but the abbreviations used by the authors are ambiguous. The authors should consider giving each complex a unique abbreviation used when in the text, figures and tables.

We appreciate this suggestion for improving clarity. We adopted a nomenclature around the intermediates (pre-TS1, TS1, post-TS1, and TS2). We consolidated the nomenclature in the text, figures and table.

4. The authors use the symbol '>' repeatedly in different contexts but never for the correct meaning of 'greater than'. This should be corrected because it is confusing.

We removed the '>' symbol throughout the revision.

5. Figures 1d, 1h are very difficult to understand. There is a mismatch between the colours used in the cryo-EM reconstruction (a lot of grey) and the schematic above of ARIH1. The implication is the grey domains in the ARIH1 schematic are grey in the reconstruction, but this presumably is not the case because other parts of the EM reconstruction correspond to SCF. The EM reconstruction figure 1d, 1h (and others) need much more careful labelling.

To address this suggestion, and to incorporate the new structural data obtained in response to Reviewer #2, we completely re-made figure 1 and eliminated these panels.

6. Fig. 3d: labelling unclear. RTI helix should be labelled.

We labeled the RTI helix (Figure 2c in the revision).

7. Fig. 4b. All subunits should be labelled.

We labeled all subunits (Extended Data Figure 7b in the revision).

8. Where is the ubiquitinase module defined?

In the revised manuscript, we added a description of the ubiquitin transferase module. This is on page 7 of the text.

9. Page 4. '.. penultimate Gly75..' doesn't make sense. It is either '.. penultimate Gly..' or '.. Gly75...'

We changed this to penultimate Gly.

10. Page 5. ARIH1* isn't defined.

We now define ARIH1*.

11. Page 5. Figure 1i (twice) doesn't exist, presumably Figure 1h.

We thank the reviewer for pointing this out. Due to adding much new data to the revision, we remade Figure 1 and updated the text.

12. Page 6. What does 'evolving catalytic configurations' mean? Not in the genetic sense presumably or is it?. The use is ambiguous.

We replaced this with: "ubiquitin's progressively changing linkage site during transfer from E2 to RBR E3 to substrate"

13. Page 6. Reference needed for the statement in the first line.

For clarity, this sentence was removed from the revision.

14. Page 8. I can't see p27 in Figure 2b.

For clarity, this figure panel was entirely re-made for the revision.

Referee #2 (Remarks to the Author):

In the manuscript, "Ubiquitin ligation to F-box protein-bound substrates by transient RING-RBR E3-E3 ligase superassembly", Schulman and colleagues provide structural details of how CRL-ARIH E3-E3 ligases mediate ubiquitin transfer to targets. CRLs represent the largest fraction of known E3 ligases and play essential roles in many areas of biology. In 2016, this group (Scott et al.) identified ARIH RBR E3 ligases as important components of some CRL ubiquitination reactions through forming CRL-ARIH E3-E3 ligase superassemblies. However, many questions remained from this work, including how these large complexes with many parts (some seemingly redundant) coordinate ubiquitin transfer from E2 to target. Notably, this work also follows up on a recent publication from the Schulman group this year (Baek et al.) that again defined structural details of how a "specialist" CRL-E2 assembly mediates ubiquitin transfer to substrates. This current study essentially marries these two previous studies through a series of impressive cryo-EM structures that track ubiquitin transfer through the enzymatic reaction from E2 -> RBR -> target. Two transition state mimics are captured that allow understanding of this process and how large structural rearrangements are induced.

Although this work is a masterpiece in structural biology,

We are very pleased by the Reviewer's enthusiasm for our work!

it is not clear to this reviewer how much closer it brings us to understanding the central questions that have been around ever since the '2 E2' model was first suggested and were accentuated by Scott et al. Below are three key questions related to this.

This comment was helpful in making us realize that our original manuscript did not adequately convey the novelty or importance of our work. Most importantly, while prior studies suggested that different ubiquitin carrying enzymes work with CRLs, it was simply not possible to envision how a CRL employs an RBR E3 to mediate ubiquitylation as this is structurally unprecedented.

Even if one had tried, models would have been inaccurate because the neddylated SCF and RBR structurally remodel each other in the amalgamated E3-E3. Moreover, the E3-E3 structure changes during the ubiquitylation cycle. At a conceptual level, our study now shows for the first time:

- How transient amalgamation of two different types of individual E3s into a neddylated SCF RING-ARIH1 RBR complex activates ubiquitin ligase activity.
- The neddylated SCF-ARIH1 RBR complex adopts radically different conformations for the two transition states.
- Both transition state configurations depend on portions of both E3s sculpting each other into active conformations unattainable by either E3 on its own, thus showing that SCFs and ARIH1 co-evolved to work with each other.
- The structural mechanism of ARIH1 RBR E3-catalyzed ubiquitylation of F-box protein client substrates is compatible with the fundamental property of CRLs: ubiquitylation of structurally diverse substrates recruited to the assembly by various receptor subunits. Our revised manuscript shows that this differs from the previously described, conventional cullin-RING-E2 structural mechanism with UBE2D-family E2s, which is optimal for a subset of SCFs and client substrates, and is incompatible with others.
- In addition to informing how the large family of SCFs ubiquitylate substrates, as the first structure showing a ubiquitylation cycle of any RBR E3, the structural data establish a foundation for understanding how a major fraction of all E3 ligases catalyze ubiquitylation.

Major points to be addressed by the authors:

- 1) What, if anything, determines whether an SCF attempts to go down the trajectory of direct transfer from E2 to substrate or via ARIH1? Is it completely stochastic? For example, does an SCFb-TrCP sometimes engage ARIH1 for priming? Does SCFSkp2 bound to p27 sometimes first recruit an E2~Ub but then the E2~Ub dissociates before it can manage to discharge its ubiquitin and priming does not occur until ARIH1 is recruited? The authors hint at potential “specialist” CRLs that may not rely on ARIH based on interaction between E2 and substrate adaptor. This includes their recent study (Baek et al.) on SCFb-TrCP-UBE2D. However, providing more generalizable insights into these questions would resoundingly improve the impact of the study.

To address these questions (as well as to address comment #1 from reviewer 1), we have performed new analyses and experiments. First, we compared the structures representing the two different mechanisms for neddylated SCF substrate ubiquitylation, which revealed striking differences (Figure 3, Extended Data Figure 7). The E3-E3 structures show the ARIH1~ubiquitin active site localized near substrates of diverse F-box proteins (SKP2 and FBXW7). Moreover, modeling another F-box protein (FBXL3) based on our cryo EM data showed its substrate, CRY1 could likewise approach the ARIH1~ubiquitin active site. The structural modeling suggested the E3-E3 mechanism is compatible with a range of geometries, including substrates proximal to an F-box protein, and also a ubiquitylated substrate. In contrast, models based on the prior structure representing ubiquitin transfer from the E2 UBE2D to a neddylated SCF^{β-TRCP}-bound peptide substrate showed that the RBX1 RING activated UBE2D~ubiquitin active site is optimally placed for substrates of β-TRCP, in part due to UBE2D directly contacting β-TRCP. Moreover, the location of the UBE2D~ubiquitin active site appeared compatible with some substrates but not others.

Second, to quantitatively compare the E3-E3 and E3-E2 mechanisms, we determined kinetic parameters by rapid quench-flow methods (Figure 3, Extended Data Table 2). This confirmed broad utility of the E3-E3 mechanism, with relatively similar catalytic efficiencies for all tested neddylated SCFs and substrates, including chain elongation of a ubiquitin-modified substrate. The conventional E3-E2 mechanism with UBE2D was optimal for neddylated SCF^{β-TRCP} and could ubiquitylate disordered peptide substrates of sufficient length to simultaneously engage an Fbp and the UBE2D~ubiquitin active site. However, ubiquitylation of substrates that cannot accommodate this E3-E2 catalytic geometry – either due to the substrate’s folded structure, or to limited distance between an F-box protein binding phosphodegron and acceptor lysine was only quantifiable for ARIH1.

- 2) How do the structural data known to date explain what appears to be a very strong tendency of ARIH1 to drive monoubiquitination? It would appear that once one ubiquitin is transferred, there is significant preference for a chain-extending E2 to take over (like Ube2R1/2 or Ube2g1/g2). Again, is there anything in the structural biology that yields clues as to how this might work, or is it completely stochastic and there is no mechanism to favor whether ARIH1 or E2 is recruited for either a naked or ubiquitylated substrate, and the system relies entirely on fast dynamics such that the residence time of ARIH1+Ube2D3~Ub or E2~Ub on Cul1-RING is short enough that the negative effects of competitive binding are negligible?

We thank the Reviewer for excellent points and questions. In fact, to our knowledge, the ability of ARIH1 to extend ubiquitin chains has not been directly tested. Thus, we included in our kinetic analyses a ubiquitin-linked substrate (ubiquitin’s C-terminus covalently bonded to acceptor residue in Cyclin E phosphopeptide via sortase-mediated transpeptidation). Indeed, ARIH1 can extend a polyubiquitin chain as well as UBE2D3 can. However, the kinetic efficiencies for both these enzymes are ~160 and ~3,400-fold lower than published kinetic parameters for ubiquitin chain elongation by UBE2R2.

Also, comparing our new structures of neddylated CRL-ARIH1 E3-E3 complexes with the previous structure showing ubiquitylation with UBE2D showed that both ubiquitin carrying enzymes engage the same surfaces of CUL1’s linked NEDD8 and RBX1’s RING, albeit in different orientations relative to the remainder of the SCF. Furthermore, the ARIH1 binding site on RBX1 was previously shown to interact with a UBE2R E2 (Spratt et al., 2012). Thus, the structural data suggested that an “active” E3-E3 would exclude these other ubiquitin carrying enzymes. We tested this by asking how ARIH1 alone (which presumably substantially adopts the autoinhibited configuration in the absence of UBE2L3~UB at equilibrium) or stable proxy for the active E3-E3 (the post-TS1 intermediate) affect substrate ubiquitylation by UBE2D3 and ubiquitin chain elongation by UBE2R2. A stoichiometric amount of the post-TS1 ARIH1 intermediate potently inhibited both UBE2D3- and UBE2R2-catalyzed ubiquitylation reactions but ARIH1 on its own was not inhibitory. The results suggest that if ARIH1 is employed by a neddylated SCF, it blocks other ubiquitin carrying enzymes; after a substrate is modified, ubiquitin-free ARIH1 would disengage, allowing a neddylated SCF to employ a different ubiquitin carrying enzyme, potentially the one with superior kinetic properties for modifying the particular Fbp client substrate (or ubiquitylated substrate).

These new data are shown in Figures 3e, Extended Data Figure 7f-g, and Extended Data Table 2.

- 3) Whereas the work shown provides detailed insight into the structures of the sequential transition states, it has much less to say about how the system traverses from the first transition

state to the second. One simple explanation is that the first transition state has pent-up potential energy, analogous to a compressed spring, and the instant that the bond linking Ub to Ube2D3 is severed, the system relaxes towards the second transition state. With this in mind, there is one analysis that I would have liked to see, which is to examine the structures using simpler traps that link the ubiquitin to either Ube2D3 or ARIH1. The latter would be particularly interesting with respect to the hypothesis above. The expectation is that the SCF+ARIH1+Ube2D3~Ub structure would look like TS1 whereas the SCF+ARIH1~Ub+Ube2D3 structure might look like TS2 (presumably the Ube2D3 would be absent).

We appreciate the Reviewer's suggestion. And the Reviewer's expectation was correct!

Specifically, we obtained a map of a "pre-TS1" intermediate: substrate-bound neddylated SCF^{FBXW7}+ARIH1+UBE2L3~Ub, where ubiquitin's C-terminus is isopeptide bonded to a lysine replacement for UBE2L3's catalytic cysteine (i.e., essentially the stable mimic published by Hay in Nature, 2012 and now commonly used to obtain structures of RING E3-E2~Ub intermediates) at 4.5 Å resolution. We also obtained 3D reconstructions of the complex representing the "post-TS1" intermediate.

The pre-TS1 intermediate does look like the TS1 intermediate, with an additional interesting feature: the Ub-guided helix and Rcat domain are not visible, presumably due to heterogenous localization within this complex.

We were also able to obtain data for a stable proxy for this highly reactive "post-TS1" intermediate, through neddylated SCF-dependent reaction of ARIH1's Cys^{cat} with the synthetic electrophilic probe ubiquitin-vinyl methyl ester (UbVME). This enabled obtaining stable mimics of the requested complex: substrate-bound neddylated SCF^{FBXW7} complex with ARIH1~ubiquitin and UBE2L3. While this sample was heterogenous, but we were able to obtain 3D reconstructions for three major classes.

In all three classes, the majority of the SCF and the entire E3-E3act superdomain are resolved. As predicted by the Reviewer, one class looks like TS2, albeit at low (12 Å overall) resolution (while UBE2L3 is not visible in this class, it cannot be excluded that UBE2L3 has dissociated from the complex wherein the E3-E3 platform is also not visible). The most abundant class refined to 7.2 Å resolution, but the only region of the ARIH1~ubiquitin E3 that is visible is the Ariadne domain within the E3-E3act superdomain. The E3-E3 platform bound to UBE2L3 is visible in only one class, but in this class the ubiquitin transferase module is not visible.

To address the Reviewer's suggestion through inclusion of these new findings, we have re-written the text and revised the figures.

Minor points:

- 1) Some of the figures may be difficult to interpret by a naïve audience. Addition of some models at key points may be helpful.

Based on suggestions from all reviewers to improve clarity, and to incorporate the new data, we have revised every main text figure and a majority of extended data figures too.

- 2) Figure 1h is referred to as Figure 1i in the text.

Figure 1 was completely re-made and panel h/i is no longer referred to in the text.

Referee #3 (Remarks to the Author):

The manuscript entitled "Ubiquitin ligation to F-box protein-bound substrates by transient RING-RBR E3-E3 ligase superassembly" by Horn-Ghetko et al., presents structures for several reconstituted complexes representing key intermediates in the transfer of ubiquitin from an E2 to an RBR E3 and from an RBR E3 to substrates bound to various flavors of reconstitute Cullin ring ligase complexes. The topic is timely, and it validates several hypotheses that were generated several years ago when it became clear that the ARIH RBRs cooperated with CRLs and NEDD8 to enhance substrate ubiquitination. The structures are all at modest resolution, but as many of the components have structures available through other means, the authors assemble a convincing array of models depicting nearly every step in this process. These structures show an array of interactions involving mutual exclusivity as well as composite surfaces that nicely illustrate mechanisms that alleviate inhibition or enforce directionality and specificity. And models explaining the dependence on Nedd8 (or lack thereof) are striking and also of significant note. This could not have been achieved without the clever use of several chemical strategies to bring normally reactive components together into stable complexes. While several of the strategies to link components together have some limitations with respect to spacing/chirality, it is the opinion of this reviewer that they are sufficient to trap relevant conformational intermediates in this process. The work is buttressed by interpretation of prior or new mutational and biochemical analyses, and thus it represents a more complete body of work in comparison to many recently published cryoEM studies. There are only a few minor concerns that should be easily addressed by the authors.

We are very pleased by the Reviewer's enthusiasm for our work!

If the authors have collected any more data in the interim, they are encouraged to improve the overall and local resolutions for the more important and relevant complexes. If this data does not exist, then the story should be published as is.

Again, we thank the reviewer for enthusiasm for our study. Our revision includes an improved local resolution map for the TS1 intermediate (3.6 Å resolution), as well as four new maps for the pre-TS1 and post TS1 intermediates (one map for pre-TS1 at 4.5 Å resolution, and three maps for post-TS1 at 7.2, 8.1, and 12 Å resolution).

One major strength of this study is the expert use of chemical strategies to bring the various reactants together, yet electron densities and corresponding atomic models for the respective linkages are either not presented or hidden behind less interesting parts of the structure (Ext Fig 3D, 3H). Close-up views of the linkages and active sites must be shown. In addition, it is difficult to ascertain the respective limitations of the chemical crosslinking strategies with respect to chirality or the relative distances between active sites or Ub tail in linked or theoretical transition states. As stated earlier, it is clear that these strategies worked to bring the reactants together, but they may not capture all of the requisite elements of the reactions if the length or topologies differ from the true transition states. For instance, Fig 1b suggests that the distance between the E2 and E3 is greater in the mimic, but by how much? Would the interface between the E2 and E3 differ if it were shorter? The reader can do these calculations, but perhaps a S to S distance could be included? Similar point in Fig 1e, is the NH of the substrate indicating the primary amine of the N-terminus or a lysine side chain? If it's the latter than the distance between the substrate, Ub and E3 is quite a bit longer in the native reaction.

To address these suggestions, we have revised Figure 1 and Extended Data Figures 1 and 2 to include these geometric details. Our revised manuscript includes a side-by-side comparison of the chemical structures for the native intermediates and stable proxies.

We designed our TS1 and TS2 probes to match the atomic distance between ubiquitin and the RBR active site during the transition states, rationalizing that the distance between ubiquitin and E2 in the TS1 probe is in the same range as that in many previous structures employing isopeptide-bonded mimics of E2~ubiquitin intermediates, and that the flexibility of peptide substrates should enable achieving an overall configuration representing the TS2 intermediate. Notably, the structures are consistent with prior extensive mutagenesis of ARIH1.

In response to this Reviewer questions, and comment 2 of Reviewer #2, we also performed additional experiments to validate the TS1 probe, which are shown in Extended Data Figure 2d.

Fig 2 d,e could be improved, the overlay makes it difficult to see what is going on. Side by side might be clearer.

We thank the Reviewer for this suggestion. We agree, and now show the structures side-by-side in the revised Figure 2.

Terms such as 'immense' and 'massive' are subjective and should perhaps be removed from the abstract.

We removed these terms.

Reviewer Reports on the First Revision:

Referees' comments:

Referee #1 (Remarks to the Author):

The authors have performed a wide range of additional experiments to address my questions. The manuscript is an important study that now warrants publication in Nature.

Referee #2 (Remarks to the Author):

The response of the authors to my initial comments has exceeded my expectation. This is yet another remarkable paper from Schulman and colleagues on the mechanism of action of SCF in collaboration with ARIH1.

I have a few specific comments, all of which can be addressed by text edits.

1. On page 2 the authors state, "Errant ubiquitylation is normally prevented by E3 ligase inhibition". This seems like a rather sweeping statement that isn't particularly well-justified based upon what is currently known. There is robust evidence of CRL inhibition by deneddylation and CAND1 and APC inhibition as well but less so for the many other ligases that have been less studied
2. On page 2 the authors state: "Dysregulated E3 ligase targeting underlies many pathologies, including microbial infections, cancers, and neurodegeneration. When I hear "dysregulated" I tend to think of gain of function/neomorphism. i.e. something becomes uncoupled. Most of the mutations that result in pathologies are due to simple loss or reduction of function. It might be more straightforward to state that reduction of E3 ligase activity underlies many pathologies.

3. On page 3 the authors state: At the distal end of the C/R domain, flexible linkers dynamically tether RBX1's E3 ligase RING domain, and CUL1's C-terminal WHB domain and its covalently linked NEDD8^{11,12}. Under a given cellular condition, dozens of structurally distinct Fbps are assembled into NEDD8-activated SCFs, the repertoire determined by substrates marked for ubiquitylation¹³⁻²⁰." The paper which perhaps best establishes the repertoire of SCFs being referred to is Reitsma et al 2017.

4. One Page 10 the authors state: "Substrate binding to its Fbp triggers SCF neddylation." I find this somewhat (unintentionally) misleading. Substrate favors the neddylated state by repressing deneddylation. However, the statement, "triggers SCF neddylation" conjures up (at least to me) a more direct role for substrate in actually promoting Nedd8 conjugation. i.e. it has a mechanistic implication that is not warranted by the available data.

Referee #3 (Remarks to the Author):

The authors adequately addressed my comments, and in my opinion, those of the other reviewers by clarifying several of the figures, by improving (simplifying) some descriptions, and by inclusion of new biochemical data that support some of the proposed models.

Author Rebuttals to First Revision:

General response to reviewers:

We are very pleased by the extremely enthusiastic responses from Reviewers! It is extremely gratifying to receive such supportive comments about our work. We are very happy that the reviewers recommend publication in Nature!

Referees' comments:

Referee #1 (Remarks to the Author):

The authors have performed a wide range of additional experiments to address my questions. The manuscript is an important study that now warrants publication in Nature.

We are very pleased by the Reviewer's enthusiasm for our work!

Referee #2 (Remarks to the Author):

The response of the authors to my initial comments has exceeded my expectation. This is yet another remarkable paper from Schulman and colleagues on the mechanism of action of SCF in collaboration with ARIH1.

We are very pleased by the Reviewer's enthusiasm for our work!

I have a few specific comments, all of which can be addressed by text edits.

1. On page 2 the authors state, "Errant ubiquitylation is normally prevented by E3 ligase inhibition". This seems like a rather sweeping statement that isn't particularly well-justified based upon what is currently known. There is robust evidence of CRL inhibition by deneddylation and

CAND1 and APC inhibition as well but less so for the many other ligases that have been less studied

This sentence was removed during the streamlining of our text to comply with Nature formatting requirements.

2. On page 2 the authors state: “Dysregulated E3 ligase targeting underlies many pathologies, including microbial infections, cancers, and neurodegeneration. When I hear “dysregulated” I tend to think of gain of function/neomorphism. i.e. something becomes uncoupled. Most of the mutations that result in pathologies are due to simple loss or reduction of function. It might be more straightforward to state that reduction of E3 ligase activity underlies many pathologies.

This sentence was removed during the streamlining of our text to comply with Nature formatting requirements.

3. On page 3 the authors state: At the distal end of the C/R domain, flexible linkers dynamically tether RBX1’s E3 ligase RING domain, and CUL1’s C-terminal WHB domain and its covalently linked NEDD8^{11,12}. Under a given cellular condition, dozens of structurally distinct Fbps are assembled into NEDD8-activated SCFs, the repertoire determined by substrates marked for ubiquitylation¹³⁻²⁰.” The paper which perhaps best establishes the repertoire of SCFs being referred to is Reitsma et al 2017.

During the revision, we rewrote the sentence and added the reference to Reitsma et al 2017 (now reference 26).

Under a given cellular condition, dozens of structurally distinct Fbps are incorporated into neddyated SCFs, the repertoire determined by substrates marked for ubiquitylation in an intricate assembly pathway¹⁹⁻²⁷.

4. One Page 10 the authors state: "Substrate binding to its Fbp triggers SCF neddylation." I find this somewhat (unintentionally) misleading. Substrate favors the neddyated state by repressing deneddylation. However, the statement, "triggers SCF neddylation" conjures up (at least to me) a more direct role for substrate in actually promoting Nedd8 conjugation. i.e. it has a mechanistic implication that is not warranted by the available data.

This sentence was removed during the streamlining of our text to comply with Nature formatting requirements.

Referee #3 (Remarks to the Author):

The authors adequately addressed my comments, and in my opinion, those of the other reviewers by clarifying several of the figures, by improving (simplifying) some descriptions, and by inclusion of new biochemical data that support some of the proposed models.

We are very pleased by the Reviewer's enthusiasm for our work!